# ALIGNING AT THE SOURCE: STEERING CORRECTIVE TO THE ORIGINS OF HARMFULNESS IN LLMS

## ABSTRACT

Persistent vulnerabilities in safety alignment hinder the deployment of large language models (LLMs). Existing methods remain susceptible to jailbreak attacks, suggesting a fundamental, unaddressed flaw in current safety paradigms. In this work, we diagnose the root cause of this fragility, identifying a systemic issue we term Depth-wise Alignment Discrepancy. We find a fundamental misalignment: harmful vectors—latent representations predisposed to unsafe content—predominantly originate in the model's lower layers, yet conventional alignment training concentrates its corrective gradients disproportionately on the topmost layers. This creates a brittle, "end-of-pipe" defense that is easily bypassed. To address this discrepancy, we propose SAGA, a framework that achieves robust safety through two synergistic innovations. First, it leverages high-entropy Chain-of-Thought (CoT) augmented data to provide the deep semantic signals necessary to reach the source of harmfulness. Second, it introduces a novel Synergistic Gradient Scaling (SGS) mechanism to explicitly reshape the gradient flow, ensuring these corrective signals are precisely applied to the identified vulnerable layers. Extensive experiments on five LLMs against six distinct jailbreak attacks demonstrate SAGA's superiority, reducing attack success rates (ASR) by 21%–63% compared to state-of-the-art baselines. Our method preserves downstream task accuracy while introducing minimal computational overhead (<3%).

## 1 INTRODUCTION

The deployment of large language models (LLMs) faces a critical challenge: while they possess immense capabilities, their safety alignment remains persistently fragile (Zou et al., 2023a). Despite extensive efforts in safety alignment, such as reinforcement learning from human feedback (RLHF)(Ouyang et al., 2022), safedecoding(Xu et al., 2024), existing methods remain vulnerable to sophisticated jailbreak attacks (Chao et al., 2023). This persistent fragility suggests that current alignment techniques fail to address a systemic structural flaw. This motivates our work to move beyond "shallow alignment"(Qi et al., 2024a) and investigate the model's internal computations to uncover the root cause of alignment fragility.

Our analysis reveals that successful jailbreaks are not merely output-layer failures but symptoms of a systemic issue rooted in the model's intermediate computations. We observe the formation of harmful vectors, which refer to the hidden state vector predisposed to decoding into harmful content, and the gradient in alignment training. We identify a core vulnerability mechanism we term **Depth-wise Alignment Discrepancy**: a fundamental misalignment between the genesis of harmful vectors and the application of alignment corrections. Specifically, we find that harmful vectors predominantly emerge in the lower layers of the model, yet conventional alignment training disproportionately concentrates its gradient-based corrections on the top-most layers. This creates a critical mismatch: the lower layers where harmfulness remains undertrained for safety to eliminate, while the top-most layers bear the entire corrective burden. This results in a superficial, "end-of-pipe" defense that is inherently brittle and easily bypassed.

To address the Depth-wise Alignment Discrepancy, the central challenge is to eliminate harmful vectors at their source. Our investigation explores a promising direction: using Chain-of-Thought (CoT)(Wei et al., 2022) augmented data to guide a deeper gradient distribution. Unlike simple refusal responses, CoT data externalizes the safety reasoning process, providing richer semantic supervision

and thus inducing stronger, more distributed learning signals (Zhang et al. 2025, OpenAI et al., 2024). Initial experiments validate this hypothesis: CoT-augmented data significantly suppresses the harmfulness peak in the lower layers. Moreover, we discover a strong positive correlation between the reasoning complexity of the CoT data—proxied by information entropy—and the magnitude of this suppression. However, while CoT provides the necessary deep signal, it is a coarse-grained instrument, insufficient to precisely match the corrective gradients to the harmfulness distribution. This leaves a residual discrepancy, necessitating a fine-grained, optimization-level control mechanism.

To this end, we propose Source-guided Alignment(SAGA), a framework designed to directly address the Depth-wise Alignment Discrepancy through a synergistic, two-pronged strategy. First, it construct high-entropy CoT dataset to serves as the semantic foundation for deeper training gradient. Second, it introduces a novel Synergistic Gradient Scaling (SGS) mechanism, which explicitly reshapes the gradient flow to target layers identified as sources of harmfulness. This synergy of data-centric depth and optimization-centric precision allows for the targeted elimination of harmful vectors generation across all layers.

The primary contributions of this work are threefold. First, we formally identify and empirically validate the Depth-wise Alignment Discrepancy as a fundamental vulnerability in LLMs. Second, we propose SAGA to address this vulnerability through a synergistic, two-pronged strategy. Finally, through extensive experiments on five LLMs against six distinct jailbreak attacks, we demonstrate that SAGA reduces Attack Success Rates by 21%–63% compared to SOTA baselines, while preserving downstream task performance and incurring negligible (<3%) computational overhead.

## 2 OBSERVATIONAL EXPERIMENTS

This section presents a series of observational experiments that systematically diagnose the root cause of alignment fragility in LLMs. We first observed the "Depth-wise Alignment Discrepancy" (section 2.1) and then explore the potential of using Chain-of-Thought to mitigate it (section 2.2).

Before presenting our observational findings, we first outline the common experimental setup and define the key metrics used throughout this section. The detailed setup is shown in appendix B.

**Models and Datasets:** Our experiments are conducted on various LLMs: Llama-3-8B(AI@Meta, 2024), DeepSeek-R1-Distill-Qwen-7B(DeepSeek-AI, 2025), and Qwen2.5-7B(Yang et al., 2024), to ensure our findings are not model-specific. To probe their alignment, we utilize two primary types of inputs: (1) a set of standard harmful queries advbench(Zou et al., 2023b), and (2) the same queries augmented with jailbreak prompts from the PAIR(Chao et al., 2023).

**Metrics:** 1)**Attack Success Rate (ASR)** is automatically judged by GPT-4. A lower ASR indicates higher robustness. 2)**Harmful Vector Distribution**: This metric is utilized to diagnose the origin of harmfulness. A "harmful vector" is defined as a hidden state that, if decoded directly from that layer, shows a higher probability of continuing with harmful content versus safe refusals, details in appendix D. 3) **Safety Key Neuron**: The neurons contribute the most to safety-critical tasks, which is computed by NA-CIA(Chen et al., 2024). The detailed is provided in Appendix E.

### 2.1 EVIDENCE: THE PHENOMENON AND CONSEQUENCES OF DISCREPANCY

This section presents a series of observational experiments designed to systematically diagnose the root cause of alignment fragility in LLMs, and proposed **Depth-wise Alignment Discrepancy**. The investigation proceeds in three stages: 1)We identify the layer-wise genesis of harmful vectors against jailbreak attacks (Section 2.1.1). 2) We visualize the stark disparity between this harmful vector distribution and the distribution of alignment-training gradients (Section 2.1.2). 3)We conduct a controlled intervention to establish a causal link between gradient allocation and alignment robustness (Section 2.1.3), thereby providing robust empirical evidence for the Depth-wise Alignment Discrepancy.

#### 2.1.1 JAILBREAK INDUCES HARMFUL VECTOR IN LOWER LAYERS

To understand the internal mechanism of a successful jailbreak, we visualize and compare the layer-wise dynamics of harmful vector formation when models respond to general harmful queries versus sophisticated jailbreak attacks.

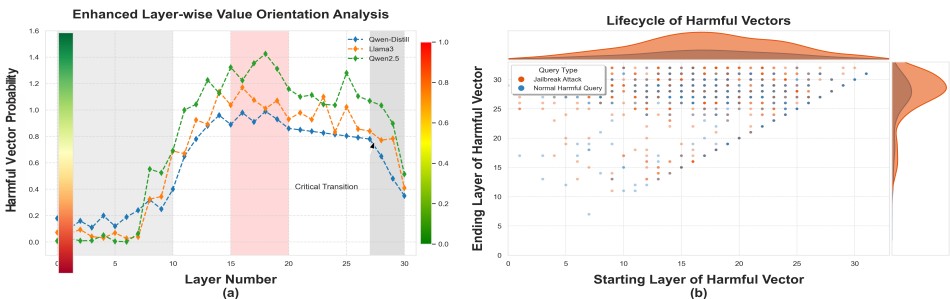

Figure 1: (a)Jailbreak induces the hidden vector distribution in lower layers. (b)Lifecycle of harmful vectors shows that the correction of harmful vectors mainly occurs at the top layers.

**Results & Analysis.** As illustrated in Figure 1, our analysis reveals a critical pattern: sophisticated jailbreak attacks induce a significant concentration of harmful vectors originating in the lower layers(10-20). While some of these vectors are corrected in the topmost layers(27-31), a substantial portion persists through to the output, successfully break the alignment. This observation directly contradicts the assumption that jailbreaks merely exploit superficial, output-layer vulnerabilities. It raises a pivotal question: if harmfulness originates deep within the model, why do existing alignment techniques fail to neutralize it at its source?

### 2.1.2 VISUALIZING THE DISCREPANCY: ALIGNMENT GRADIENTS VS. HARMFULNESS

To explore solutions to the above issues, we observed the training process and discovered that the mismatch between where alignment learning occurs and where harmful vectors originates.

**Results & Analysis.** We calculate the Harmful vector distribution pointing "the locus of harmful vector" and Safety neuron distribution as well as training gradient distribution pointing "the locus of alignment learning". Figure 2(a) reveals a severe spatial incongruity. The generation of harmful vector peaks in the middle layers (start approx. 15-20). Conversely, the training-gradient escalation occurs in layers posterior to the surge of harmful vectors(13 vs 8) and is concentrated at the topmost layers(28-31). This creates a **depth-wise alignment discrepancy**: alignment training corrections are disproportionately applied far from the source of the problem. Figure 2(b) further validates that the reduction in harmful vectors layer-by-layer is positively correlated with the applied gradient magnitude. We also have performed a formal statistical analysis, the result (Spearman's $= 0.87$ ,$p < 0.001$) provides strong statistical evidence of this monotonic positive correlation, providing rigorous, quantitative proof beyond visual intuition. The consequence of this mismatch is an inefficient and brittle "end-of-pipe" defense, where the model expends its corrective capacity on suppressing symptoms at the output layer, rather than eliminating the root cause in the intermediate layers. This superficial correction is, unsurprisingly, easily bypassed by targeted attacks.

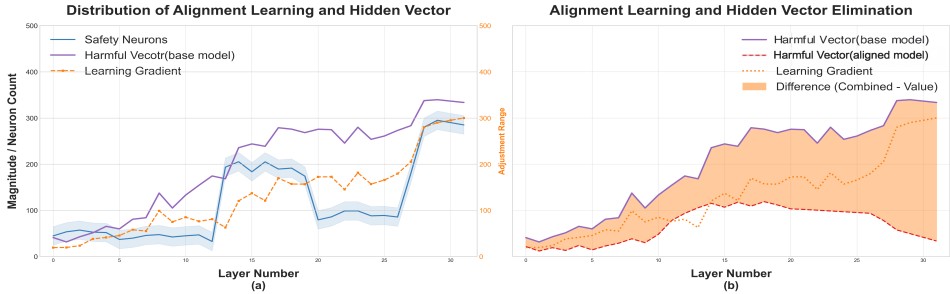

Figure 2: (a)The visualization of depth-wise alignment discrepancy. (b)Reduction in harmful vectors layer-by-layer is positively correlated with the applied gradient magnitude.

### 2.1.3 CAUSAL LINK: HARMFULNESS REDUCTION VIA TRAINING GRADIENT

While the correlation observed above is strong, we now seek to establish a direct causal link between the depth of depth-wise alignment discrepancy and the model's final alignment robustness, as well as confirm that the observed vulnerability is indeed caused by shallow gradient distribution.

We conducted a controlled experiment using RLHF on a single base model. In a series of independent training runs, we froze the entire model except for the final N layers, where N was set to 1, 5, 10, 20. This setup allowed us to precisely control the depth to which alignment gradients could penetrate the network. We then evaluated the final Attack Success Rate (ASR) of each resulting model on a held-out set of jailbreak prompts.

**Results & Analysis.** The results, presented in Figure 3, unequivocally demonstrate a causal relationship. Fine-tuning only a few top layers (N=1, 10) yields negligible improvements in ASR, confirming that such shallow corrections are ineffective. The model's robustness improves meaningfully and monotonically only as N increases, allowing alignment gradients to reach the deeper, vulnerability-prone middle layers. Figure 3(a) visualizes how a larger N allows for the elimination of harmful vectors at deeper layers, and the extent of this eradication remains positively correlated with the training gradient throughout. Figure 3(b) shows the direct impact on the final ASR. This causally validates our central thesis: a robust alignment requires the targeted elimination of harmful vectors at their source, which is vulnerable when corrective gradients are confined to the topmost layers.

Collectively, these three experiments provide robust, multi-faceted evidence for the Depth-wise Alignment Discrepancy. We have shown that harmfulness originates in lower layers, that standard alignment focuses on upper layers, leading to a shallow alignment training. This discrepancy is the direct cause of persistent vulnerabilities.

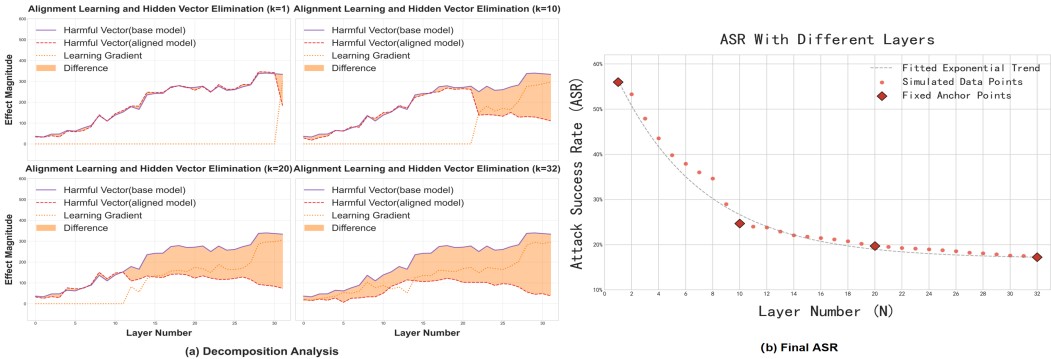

Figure 3: (a)The layer-wise gradient distribution positively correlates with the elimination of harmful vectors. (b)The fewer layers of aligned training, the higher the asr.

### 2.2 AN OPPORTUNITY TO MITIGATE DISCREPANCY: THE POWER OF SAFE REASONING

The preceding analysis established that Depth-wise Alignment Discrepancy arises from a training gradient signal that is too "shallow", leading a vulnerable safety alignment. In this section, we investigate and demonstrate that utilizing safety reasoning CoT in training data can effectively mitigate this phenomenon.

### 2.2.1 CoT'S MACRO-LEVEL IMPACT ON HARMFULNESS AND GRADIENTS

In section 2.1, we also observed that models with reasoning capabilities have a lower probability of generating harmful vectors than non-reasoning models (DeepSeek-R1-Distill-Qwen-7B vs Qwen2.5-7B). The most salient distinction between reasoning and non-reasoning models lies in the data format they generate and are trained on: a complex, multi-step chain-of-thought precedes the final response. Unlike a simple refusal response data for non-reasoning model, an extra safety reasoning process, which is formed as chain of Thought(CoT) text, requires the model to access, integrate, and articulate abstract safety principles and causal relationships—cognitive processes widely believed to be rooted

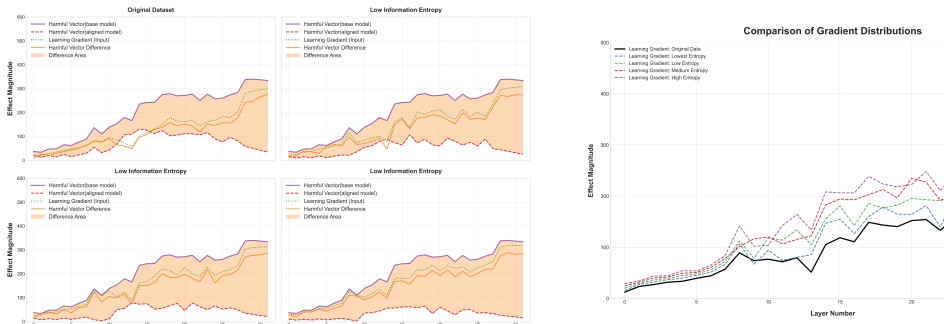

Figure 4: A monotonic increase in information entropy yields a correspondingly deeper gradient distribution and greater eradication of harmful vectors across all layers.

in the model's lower layers. This insight motivates us to take safety reasoning CoT as a starting point for exploring methods to alleviate the discrepancy.

To investigate whether safety-reasoning CoT augmented data can mitigate the Depth-wise Alignment Discrepancy by inducing a deeper gradient flow, We constructed a CoT-augmented safety dataset using the SafeChain(Ma et al., 2024) framework. To analyze the impact of reasoning complexity, we partitioned this augmented corpus into high-, mid-, and low-complexity tiers based on the information entropy of the reasoning chains. Higher entropy corresponds to more diverse and intricate reasoning trajectories. The details are shown in appendix G.

**Results & Analysis.** The results in Figure 4 confirms that CoT-augmented training provides a powerful, albeit incomplete, solution to the discrepancy problem. We observe two significant improvements: 1) Suppression of Harmfulness Origin: CoT training substantially suppresses the harmfulness peak in the lower-middle layers. The data demonstrates a strong positive correlation between the CoT's information entropy and the magnitude of this suppression; more complex reasoning leads to a more profound reduction in deep-seated harmfulness. 2) Deepening the Locus of Learning Gradient: CoT with higher entropy induces a more uniform and deeper gradient distribution compared to standard alignment. It effectively shifts the "locus of learning" from the top-most layers towards the lower layers where harmful vectors originate. The more theoretical rational analysis is shown in appendix G and F.

However, this macro-level success reveals a micro-level limitation. While the CoT-induced gradient distribution is deeper, it is not precise. The corrective signal, generated from a long and complex reasoning chain, is powerful but diffuse. As shown in the comparison, it still fails to perfectly match the specific, fine-grained distribution of harmful vectors. This leaves a residual discrepancy, explaining why harmfulness, though significantly reduced, is not entirely eradicated.

This final observation is critical. It suggests that a complete solution requires a synergistic, two-pronged approach. We have found a data-centric method (CoT) to guide gradients to the lower layers. What is now needed is a complementary optimization-centric mechanism to precisely sculpt the gradient flow within that region, ensuring corrections are applied with surgical precision. This insight directly motivates the design of our SAGA framework, which we introduce in the following section.

## 3 METHODOLOGY

Based on the insights from our observational experiments, we introduce **SAGA**, a framework designed to directly address the Depth-wise Alignment Discrepancy and instill a more resilient safety alignment. SAGA is realized through two synergistic modules: (1) a **Iterative CoT Refinement via Entropy Ascent**, which serves as the semantic foundation for deeper training gradient. By generating high-entropy, complex reasoning chains, this module provides the necessary deep learning signals that penetrate the model's lower layers. (2) **Synergistic Gradient Scaling** reshapes the gradient flow during training, ensuring that the corrective updates are precisely allocated to the harmful vector distribution. The workflow is shown in Figure 5.

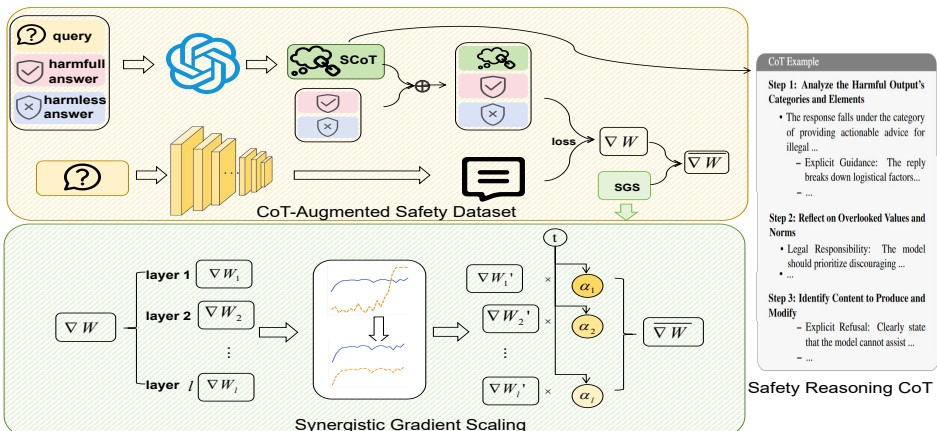

Figure 5: CoT-Augmented Dataset phase generates the datasets for subsequent training. Synergistic Gradient Scaling adjust the gradient to promote deeper and broader training.

## 3.1 ITERATIVE CoT REFINEMENT VIA ENTROPY ASCENT

Our objective is to generate not merely a plausible, but a maximally informative safety reasoning CoT for each harmful query-answer pair $(Q, A)$. We move beyond a simple generate-and-select paradigm and introduce an iterative refinement procedure, which we term *Entropy Ascent*. This process progressively enhances a CoT's complexity and informational richness by optimizing for its self-information, treating it as a direct proxy for information entropy.

The process is formalized as an iterative loop. At each step $t$, we maintain a candidate safety reasoning CoT, $C^{(t)}$.

**Initialization** ($t = 0$). For a given query-answer pair $(Q, A)$, we begin by generating an initial seed CoT, $C^{(0)}$, using a prompt with the teacher model, $\mathcal{M}_{\text{teacher}}$ (GPT-o3).

**Entropy Evaluation.** To quantify the information content of any candidate CoT $C$, we employ the **Self-Information Score**, $H_{\text{self}}(C)$. This metric, defined as the negative log-likelihood of the sequence according to the base model $\mathcal{M}_{\text{ref}}$, measures the sequence's unpredictability. A higher score signifies a less probable, and thus more informative, sequence from the perspective of the base model. For a CoT $C$ comprising $N$ tokens $(w_1, \ldots, w_N)$, the score is:

$$H_{\text{self}}(C) = -\frac{1}{N} \sum_{i=1}^{N} \log_2 p_{\text{ref}}(w_i | w_1, \ldots, w_{i-1}) \tag{1}$$

where $p_{\text{ref}}(w_i | \cdot)$ is the conditional probability assigned by $\mathcal{M}_{\text{ref}}$. The detail introduction is shown in appendix G.

**Iterative Refinement** ($t \geq 1$). The core of our method is a refinement loop that continues as long as the entropy score of the current best CoT, $H_{\text{self}}(C^{(t-1)})$, is below a predefined threshold $\tau$, or a maximum iteration count $T$ is not reached. In each iteration $t$:

1. **Meta-Prompting for Ascent:** We construct a dynamic *meta-prompt* designed to guide the teacher model toward a higher-entropy state. This prompt includes the query $Q$, the previous CoT $C^{(t-1)}$, its score $H_{\text{self}}(C^{(t-1)})$, and an explicit instruction. For instance: "The following reasoning achieved an information score of [score]. Refine it by introducing more detailed logical steps, exploring alternative perspectives, or connecting disparate concepts to increase its analytical depth and informational novelty."

2. **Candidate Generation:** Using this meta-prompt, we generate a new pool of $k$ candidate refinements $\{C_1^{(t)}, \ldots, C_k^{(t)}\}$ from $\mathcal{M}_{\text{teacher}}$.

3. **Greedy Selection:** We select the candidate from this pool that maximally increases the self-information score to serve as the new anchor for the next iteration.

$$C^{(t)} = \underset{C_i^{(t)} \in \{C_1^{(t)}, \ldots, C_k^{(t)}\}}{\arg\max} H_{\text{self}}(C_i^{(t)}) \tag{2}$$

The loop terminates when $H_{\text{self}}(C^{(t)}) \geq \tau$ or $t = T$. The final, optimized CoT is denoted as $C^* = C^{(t_{\text{final}})}$. This procedure ensures that the resulting CoT is not just selected from a static set, but is actively constructed to be information-rich. This process is repeated for all queries to yield a refined dataset, $D_{\text{refined}}$.

## 3.2 SYNERGISTIC GRADIENT SCALING (SGS)

In this section, we introduce **Synergistic Gradient Scaling (SGS)**, which reshapes the gradient distribution to align with the harmfulness distribution, thereby directly rectifying the core discrepancy. SGS is operationalized in two stages.

**Stage 1: Establishing the Target Harmfulness Distribution**    Before training, we first construct a "correction blueprint" by quantifying the base model's intrinsic propensity to form harmful representations at different layers. This results in a **Target Harmful Vector Distribution**, denoted as $\mathcal{H} \in \mathbb{R}^L$. We quantify the per-layer frequency of harmful vector emergence, $E[s_l]$, on a disjoint diagnostics dataset and normalize it to form the target distribution $\mathcal{H}$:

$$\mathcal{H}_l = \frac{E[s_l]}{\sum_{k=1}^{L} E[s_k]} \tag{3}$$

This distribution identifies the layers where the alignment problem is most acute. To ensure $\mathcal{H}$ remains relevant as the model evolves, we re-computing the distribution at the end of each training epoch. This balances target accuracy with training stability.

**Stage 2: Dynamic Gradient Adaptation During Training**    With the target distribution $\mathcal{H}$ established, SGS dynamically modulates the layer-wise gradients at each training step $t$. First, we monitor the optimizer's default behavior by computing the instantaneous gradient distribution, $A(t)$, derived from the L2 norm of the gradients for each layer's parameters, $\|\nabla \mathcal{W}_l(t)\|_2$:

$$A_l(t) = \frac{\|\nabla \mathcal{W}_l(t)\|_2}{\sum_{k=1}^{L} \|\nabla \mathcal{W}_k(t)\|_2} \tag{4}$$

The core of SGS is defining a scaling factor $\alpha_l(t)$ to bridge the gap between the target $\mathcal{H}_l$ and the actual flow $A_l(t)$:

$$\alpha_l(t) = 1 + \delta(t) \cdot \left( \frac{\mathcal{H}_l}{A_l(t) + \epsilon} - 1 \right) \tag{5}$$

where $\epsilon$ is a small constant for numerical stability. The intervention strength is modulated by a time-dependent(current epoch number as $t$) coefficient $\delta(t)$, which follows a cosine decay schedule. This allows for aggressive intervention early in training, which gradually anneals to zero for convergence. Fundamentally, this per-layer rule is designed to solve a global distributional objective: minimizing the Kullback-Leibler (KL) divergence between the target and actual gradient distributions, $D_{\text{KL}}(\mathcal{H}\|A(t))$. The resulting updated gradient distribution $A'(t)$ is a convex combination of the original and target distributions, robustly guiding the model towards a deep and resilient alignment.

The final step is to apply these scaling factors to the originally computed gradients. The modulated gradient for each layer, $\nabla \mathcal{W}_l'(t)$, is obtained by:

$$\nabla \mathcal{W}_l'(t) = \alpha_l(t) \cdot \nabla \mathcal{W}_l(t) \tag{6}$$

These modulated gradients $\{\nabla \mathcal{W}_1'(t), \ldots, \nabla \mathcal{W}_L'(t)\}$, which now embed the corrective information from our target distribution $\mathcal{H}$, are then passed to the base optimizer for the final parameter update. This mechanism ensures that layers identified as primary sources of harmfulness receive proportionally larger corrective updates, directly rectifying the Depth-wise Alignment Discrepancy within the optimization process itself.

## 4 EXPERIMENT

In this section, a series of experiments is designed to evaluate SAGA across robustness, usefulness, and efficiency. The experiment setup is shown in appendix B.

**Evaluation Metrics**: Attack Success Rate (ASR) is utilized as the metric to evaluate the alignment security. We use llama-guard (Team, 2024), and the manual review to judge the response harm assessment. For downstream task, we use accuracy (ACC) as the evaluation indicator.

### 4.1 EXPERIMENTAL RESULTS

| Model | Method | No Attack↓ | GCG↓ | AutoDAN↓ | codeattack↓ | Pair↓ | ArtPrompt↓ |
|-------|--------|-----------|------|----------|-------------|-------|------------|
| Vicuna-13B | No Defense | **0.0%** | 93.97% | 80.15% | 58.32% | 92.40% | 40.99% |
| | PPL | 8.06% | **0.0%** | 84.00% | 50.41% | 81.90% | 42.13% |
| | RLHF | 7.03% | 12.18% | 18.25% | 26.53% | 25.44% | 13.95% |
| | Self-Reminder | **0.0%** | 41.53% | 21.31% | 40.10% | 46.03% | 29.09% |
| | Retokenization | 40.85% | 67.51% | 31.97% | 50.13% | 77.14% | 36.38% |
| | AED | **0.0%** | 13.88% | 21.48% | 31.57% | 35.22% | 13.44% |
| | Safedecoding | **0.0%** | 12.03% | 27.98% | 36.52% | 10.26% | 28.25% |
| | SAGA | **0.0%** | 4.40% | 15.53% | 18.55% | 9.50% | 13.05% |
| Llama3-8B | No Defense | **0.0%** | 33.91% | 25.05% | 51.83% | 28.46% | 40.72% |
| | PPL | **0.0%** | **0.0%** | 9.45% | 40.91% | 17.01% | 29.44% |
| | RLHF | 1.12% | 3.58% | 9.42% | 18.88% | 17.75% | 31.46% |
| | Self-Reminder | **0.0%** | 2.90% | 11.35% | 39.07% | 15.74% | 29.84% |
| | Retokenization | **0.0%** | 5.93% | 10.00% | 45.12% | 11.64% | 36.54% |
| | AED | **0.0%** | 4.10% | 10.28% | 19.55% | 15.80% | 16.95% |
| | Safedecoding | 0.86% | 2.14% | 16.15% | 16.7% | **3.42%** | 15.17% |
| | SAGA | **0.0%** | 4.80% | 14.23% | 13.55% | 13.73% | 13.05% |
| Qwen2.5-7B | No Defense | 8.51% | 86.32% | 82.12% | 46.65% | 87.52% | 32.79% |
| | PPL | 6.45% | **0.00%** | 75.20% | 40.33% | 65.52% | 33.70% |
| | RLHF | 5.62% | 17.02% | 24.60% | 23.22% | 28.35% | 27.16% |
| | Self-Reminder | **0.00%** | 33.22% | 17.05% | 32.08% | 36.82% | 23.28% |
| | Retokenization | 32.68% | 53.99% | 25.58% | 40.10% | 61.71% | 29.10% |
| | AED | **0.00%** | 9.50% | 17.18% | 25.25% | 28.17% | 10.73% |
| | Safedecoding | **0.00%** | 3.28% | 10.59% | 10.88% | 18.65% | 8.06% |
| | SAGA | **0.00%** | 3.02% | 8.63% | 7.78% | 10.40% | 4.27% |
| DeepSeek-R1 | No Defense | 0.80% | 94.55% | 88.70% | 65.40% | 89.15% | 55.20% |
| | PPL | 0.50% | **0.00%** | 58.20% | 42.10% | 68.30% | 40.15% |
| | RLHF | 1.10% | 18.40% | 22.50% | 25.30% | 30.15% | 24.50% |
| | Self-Reminder | **0.00%** | 8.20% | 20.10% | 28.50% | 38.20% | 22.10% |
| | Retokenization | 6.50% | 35.40% | 30.20% | 45.20% | 62.10% | 35.40% |
| | AED | **0.00%** | 14.50% | 21.30% | 28.10% | 32.40% | 18.50% |
| | Safedecoding | 0.40% | 6.20% | 12.50% | 15.40% | 14.20% | 12.80% |
| | SAGA | **0.00%** | **3.20%** | **7.50%** | **6.80%** | **5.40%** | **4.10%** |
| QwQ | No Defense | 6.81% | 73.00% | 64.23% | 44.32% | 73.15% | 34.43% |
| | PPL | 5.56% | **0.00%** | 54.46% | 38.31% | 62.29% | 32.07% |
| | RLHF | 4.84% | 15.32% | 23.33% | 22.11% | 27.27% | 25.81% |
| | Self-Reminder | **0.00%** | 31.56% | 16.20% | 30.48% | 35.16% | 22.14% |
| | Retokenization | 29.34% | 51.34% | 24.30% | 38.09% | 58.77% | 27.65% |
| | AED | **0.00%** | 8.55% | 16.32% | 24.01% | 26.73% | 10.22% |
| | Safedecoding | **0.00%** | 3.12% | 10.12% | 10.34% | 17.78% | 7.71% |
| | SAGA | **0.00%** | 2.03% | 7.07% | 6.71% | 7.43% | 3.96% |

Table 1: The alignment performance(ASR) of applying alignment methods with various jailbreak methods. We bold the best performing.

**Robustness against Jailbreak**   Table 1 shows that SAGA achieves the lowest ASR on almost all models. Especially on large reasoning models, SAGA not only ensures the harmlessness of solutions but also maintains the lowest harmfulness during the CoT compared to other methods. This proves that SAGA significantly enhances the robustness against adversarial methods. The Figure 6 shows that harmful vectors are significantly eliminated across all layers. This provides direct evidence that SAGA fundamentally prevent the formation of harmful representations at their source. It is worth noting the exceptionally low ASR against GCG (0%). This is not an anomaly but a result of SAGA's "deliberative" safety mechanism. We provide a detailed analysis of this phenomenon and results under extended attack budgets in Appendix M.

**Robustness Against Fine-tuning**   Beyond direct attacks, a robust alignment must also withstand the parameter perturbations introduced by downstream task fine-tuning. Table 2 shows the Attack Success Rates (ASR) on five models after they have been fine-tuned on a downstream task. SAGA also consistently achieves the lowest ASR, demonstrating superior resilience against alignment collapse. We attribute this structural robustness to the fact that our method promotes the training of a broader and more distributed set of safety-critical neurons as shown in Figure 9.

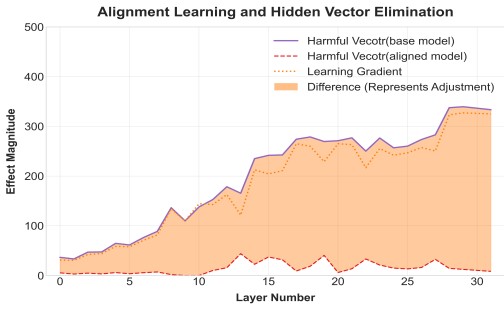

Figure 6: Our method achieves complete layer-wise eradication of harmful vectors.

| Method | Llama2-7B | Vicuna-13B | Mistral-7B | Qwen2.5-7B |
|--------|-----------|-----------|-----------|-----------|
| RLHF | 16.53% | 26.53% | 17.59% | 23.22% |
| Finetuning | 60.47% | 55.40% | 66.74% | 52.25% |
| SSAH | 21.17% | 16.62% | 20.02% | 15.67% |
| RESTA | 14.52% | 15.51% | 21.36% | 14.63% |
| SAGA | **7.26%** | **6.65%** | **8.01%** | **5.22%** |

Table 2: The alignment capability of applying alignment methods after subsequent finetuning.

**SAGA is Helpful** Figure 7 presents the impact of SAGA on LLMs' downstream task performance. The experiments indicate that SAGA achieves the highest accuracy in the three downstream tasks, with no significant changes compared to the original model. In addition, using a training dataset with CoT also enhances the reasoning ability of the model to a certain extent.

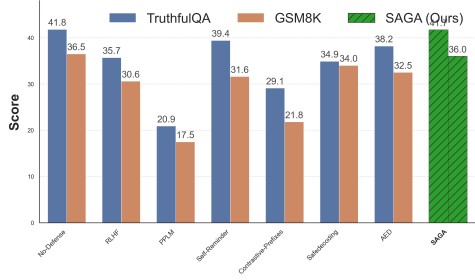

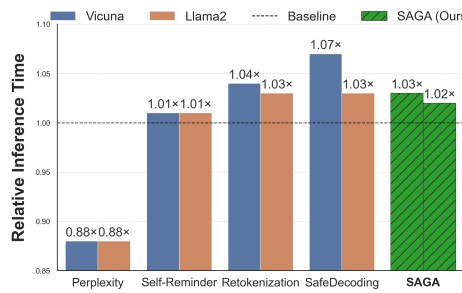

Figure 7: The generation performance(ACC) of applying protective methods

Figure 8: Comparison of inference time for Vicuna and Llama2.

**SAGA is Efficient** We compare the inference time and computational latency of SAGA with SOTA methods. Figure 8 shows that the latency of SAGA is only 3% in Llama2 and 2% in Vicuna compared to no defense, which substantiates that it does not affect the computational efficiency.

**Ablation Studies** : The result in Table 3 shows that both mechanisms of CoT -augmented data and SGS are effective and necessary. High-entropy CoT data alone cannot precisely and completely eliminate harmful vectors at every layer, whereas relying solely on SGS lacks sufficient safety knowledge and gradient signal, rendering convergence unattainable. These two mechanisms work synergistically to enhance robust safety alignment.

## 4.2 SAGA IS NOT OVER-REFUSAL

To verify that SAGA achieves safety without compromising utility, we conducted a dual-faceted evaluation focusing on inappropriate refusal rates and reasoning capabilities under sensitive framing.

**Evaluating False Positive Refusals.** As presented in Table 4, SAGA consistently achieves the lowest inappropriate refusal rates across model families. Notably, on Llama-3-8B, SAGA reduces the refusal rate on XSTest by 52% compared to the official RLHF baseline (2.3% vs. 4.8%). This indicates that our method essentially improves the precision of the safety boundary, allowing the model to distinguish between malicious intent and benign usage of sensitive vocabulary.

**Robust Reasoning in Sensitive Contexts (GSM8K-SC).** To rigorously test whether safety mechanisms interfere with complex cognitive tasks, we constructed a specialized stress-test dataset:

| Attack | SAGA | CoT-augmented | SGS |
|--------|------|---------------|-----|
| No Attack | 0.0% | 0.0% | 0.0% |
| GCG | 1.66% | 1.89% | 3.98% |
| AutoDAN | 5.95% | 6.54% | 13.23% |
| codeattack | 4.92% | 5.21% | 10.54% |
| Pair | 4.11% | 4.57% | 9.37% |
| ArtPrompt | 7.03% | 7.58% | 19.65% |

Table 3: Ablation experiment of DeepAlign. CoT-augmented represents only using the CoT-augmented data module. SGS represents only using the Synergistic Gradient Scaling Module.

| Benchmark | Model | No Defense (Base) | RLHF | SAGA (Ours) |
|-----------|-------|-------------------|------|-------------|
| OR-Bench-80k | Llama 2-7B | 16.5 | 21.6 | **15.0** |
| | Llama 2-13B | 14.9 | 18.2 | **13.3** |
| | Llama 3-8B | 6.8 | 11.5 | **6.5** |
| | Qwen1.5-7B | 4.4 | 12.7 | **4.6** |
| XSTest | Llama 2-7B | 38.4 | 40.5 | **32.5** |
| | Llama 2-13B | 32.0 | 34.7 | **28.9** |
| | Llama 3-8B | 1.9 | 4.8 | **2.3** |
| | Qwen1.5-7B | 1.4 | 5.1 | **1.4** |

Table 4: Comparison of Inappropriate Refusal Rates (%) on Benign but Sensitive Benchmarks. Lower scores indicate better utility preservation. SAGA significantly outperforms standard RLHF baselines, reducing false refusals.

**GSM8K-Sensitive-Context (GSM8K-SC).** This dataset re-contextualizes standard mathematical problems into sensitive domains (e.g., medical triage, conflict zones) while preserving the exact mathematical structure (details in Appendix K). The results in Table 5 reveal a negligible performance gap between the original GSM8K and GSM8K-SC for SAGA-aligned models. For instance, Llama-3-8B maintains a high accuracy of 79.1% on the sensitive set, virtually identical to the 79.2% on the original set. This confirms that SAGA does not indiscriminately suppress outputs based on contextual triggers.

| Model | GSM8K (Original) | GSM8K-SC (Sensitive Context) |
|-------|------------------|------------------------------|
| Llama 2-7B | 17.1 | 17.1 |
| Llama 2-13B | 29.5 | 29.3 |
| Llama 3-8B | 79.2 | 79.1 |
| Qwen-1.5-7B | 72.1 | 72.1 |

Table 5: Accuracy (%) Comparison between Standard GSM8K and GSM8K-SC. The negligible difference ($\Delta \approx 0$) confirms that SAGA maintains reasoning capabilities even when problems are framed in sensitive contexts.

**Mechanism of Utility Preservation.** We attribute this superior trade-off to the high-entropy Chain-of-Thought (CoT) data utilized in SAGA. Unlike shallow alignment methods that often rely on simple pattern matching (leading to over-refusal), the CoT signals provide the model with explicit reasoning chains about *why* a request is safe or unsafe. This fosters a principled understanding of safety guidelines, enabling the model to navigate sensitive topics with the nuance required to serve user intent while filtering out genuine harm.

## 5 CONCLUSION

In this work we observed the Depth-wise Alignment Discrepancy and propose SAGA, a novel framework that synergistically combines CoT-augmented data for deep semantic guidance with an SGS mechanism for precise control, to alleviated discrepancy and achieve a robust safety Alignment.

ETHICS STATEMENT

This research adheres to the ICLR Code of Ethics. Our work focuses on understanding and mitigating the generation of harmful, unethical, or dangerous content by Large Language Models (LLMs). The primary goal of this research is defensive: to develop more robust safety alignment techniques that can prevent the misuse of AI systems.

During our investigation, we utilized existing public datasets designed for safety research, such as AdvBench and PAIR, which contain harmful prompts. All experiments involving the generation of potentially harmful content were conducted in a controlled and isolated research environment. The outputs were used exclusively for analytical purposes to diagnose model vulnerabilities and validate our proposed defense mechanism.

We acknowledge the dual-use nature of research into model vulnerabilities. To mitigate the risk of misuse, we have deliberately refrained from releasing any successfully generated harmful outputs. Our public contributions, including source code, will be focused solely on the implementation of our defensive framework, SAGA. We believe that the potential benefits of creating safer and more reliable LLMs, thereby reducing their potential for societal harm, substantially outweigh the risks associated with this research.

REPRODUCIBILITY STATEMENT

We are committed to ensuring the reproducibility of our research. All assets required to reproduce our findings are detailed throughout the paper and its appendices.

- **Algorithms and Models:** The detailed architecture and mechanics of our proposed SAGA framework, including the CoT-Augmented Dataset generation and the Synergistic Gradient Scaling (SGS) algorithm, are presented in Section 3. A complete, open-source implementation of our framework will be made available as supplementary material and upon publication.
- **Experimental Setup:** The configurations for all experiments, including model versions, training hyperparameters (for DPO, RLHF, etc.), and specific settings for our observational studies, are documented in Appendix B.
- **Datasets:** We utilized publicly available datasets, including AdvBench, PAIR, TruthfulQA, and GSM8K. The specific subsets and preprocessing steps are described in Section 4.1. The methodology for constructing our CoT-augmented dataset is detailed in Section 3.1.
- **Metrics:** The precise mathematical formulations and calculation procedures for our key diagnostic metrics, `H_origin` and `A_align`, are provided in our new Section 2.1 and detailed in Appendix F.

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

## USE OF LARGE LANGUAGE MODELS (LLM) STATEMENT

During the preparation of this manuscript, a Large Language Model (LLM) was utilized as an advanced writing and editing assistant. The primary roles of the LLM were to: (1) refine and rephrase sentences and paragraphs to improve clarity, conciseness, and academic tone; (2) assist in restructuring sections to enhance the logical flow and narrative strength of our arguments; and (3) provide translations of technical sections between English and Chinese to ensure consistency and accuracy.

The LLM did not contribute to the core research ideation, including the formulation of the "Depth-wise Alignment Discrepancy" hypothesis, nor did it design the SAGA framework or the experiments presented. The LLM did not generate or analyze any experimental data. All scientific claims, experimental results, and conclusions were conceived and verified by the human authors, who hold full responsibility for the content of this paper.

## A  RELATED WORKS

### A.1  EXISTING WORK ON MODEL SAFETY

Many existing studies have already highlighted the vulnerability of safety alignment. Qi et al. (2024a) first proposed "Shallow Alignment" concept, which indicates that safety strategies constrained to initial tokens are easily broken by subsequent fine-tuning. Wei et al. (2024) found that safety-critical regions are very sparse, accounting for only 3% at the parameter level. SSAH(Li & Kim, 2024) suggests that safety alignment mainly focuses on a simple binary classification task of either refusing or fulfilling requests. Lee et al. (2024) reveals that harmful knowledge isn't removed during alignment training, and jailbreak attacks can elicit it. However, our work distinguishes itself from these studies by not only analyzing the intrinsic manifestations of vulnerability but also delving into its origins within the alignment training process and exploring potential mitigation strategies.

## A.2 ALIGNMENT METHODS

Existing alignment efforts have attempted to improve robustness against fine-tuning and jailbreak attacks. RESTA(Bhardwaj et al., 2024) can be re-aligned for safety during fine-tuning via mathematical operations on model parameters, restoring model safety by adding safety vectors to fine-tuned model parameters. SSAH(Li & Kim, 2024) froze 7.5% of safety-critical components during fine-tuning. Our approach differs from theirs in that we enhance the intrinsic robustness of model alignment without interfering with subsequent fine-tuning. This allows us to maintain alignment capabilities while also preserving better downstream task performance.

A critical challenge in deploying Large Language Models (LLMs) is the erosion of safety alignment during downstream fine-tuning. Safely Partial-Parameter Fine-Tuning (SPPFT)(Li et al., 2025) addresses this by identifying and freezing a contiguous block of middle layers—deemed "safety layers"—to prevent the catastrophic forgetting of refusal behaviors. In contrast, our work extends this line of inquiry with a in-depth, layer-wise analysis of safety mechanisms and the origins of harmful vectors, revealing a phenomenon we term "Depth-wise Alignment Discrepancy." Whereas SPPFT's approach is primarily preservative—seeking to maintain existing safety capabilities—our method, SAGA, leverages these granular insights to proactively construct a more fundamentally robust safety alignment.

## B EXPERIMENT SETUP

**Attack Datasets:** We utilized **Advbench**(Zou et al., 2023b) and **HEx-PHI**(Qi et al., 2024b) as attack query datasets as test datasets to validate the safety of SAGA.

**Downstream Tasks Datasets:** TruthfulQA (Lin et al., 2022) is comprised of questions that are formulated to challenge the veracity of the model's outputs, which are used to evaluate the truthfulness and reliability of the generated response. GSM8K (Cobbe et al., 2021) is aimed at evaluating the model's proficiency in understanding and solving complex mathematical problems typically encountered at the grade school level.

**Target model**:We validate SAGA on five following models: **Vicuna-13b** (Mukherjee et al., 2023), **LLaMA2-7b** (Touvron et al., 2023), **Mistral 7b** (AI, 2023), (DeepSeek-AI et al., 2024), **deepseek-r1** (DeepSeek-AI et al., 2025) .

**Baseline**: The detailed baseline settings and specific configurations for each experiment are described in the appendix C.

**SafeDecoding** (Xu et al., 2024) ensures safe and reliable outputs by applying constraints during the decoding process. **Self-Reminder** (Xie et al., 2023) involves incorporating mechanisms that prompt it to self-check or reflect on its generated responses. **PPL** (Perplexity) (Alon & Kamfonas, 2023) assesses the uncertainty in a model's output and detects potentially harmful or nonsensical responses. **RLHF** (Reinforcement Learning from Human Feedback) (Ouyang et al., 2022) refines an LLM using reinforcement learning. **Retokenization** (Jain et al., 2023) adjusts the tokenization process to modify or restrict the vocabulary or input sequences. **AED** (Liu et al., 2024) (Adversarial Example Detection) identifies and filters adversarial inputs or examples that might cause a model to behave unpredictably or maliciously.

**Implementation Details.** All models are trained for 3 epochs using the AdamW optimizer with a learning rate of 2e-5 and a batch size of 16. The hybrid loss hyperparameter $\alpha$ in ALCA is set to 0.4. Our implementation utilizes PyTorch and the Hugging Face Transformers library.

## C BASELINE SETUP

Here's the translation of your description into English, suitable for an academic setting within a research paper on LLM alignment:

Experimental Setup Supervised Fine-Tuning (SFT) For SFT, we randomly sampled 20% of the dataset for training purposes. The model was fine-tuned using the Supervised Fine-Tuning method with the following configuration:

Precision: fp16 Trainer configuration: Number of nodes: 1 Number of devices: 2 Micro batch size: 1 Global batch size: 32 Maximum sequence length: 1024 Learning rate: 1e-5 Reinforcement Learning from Human Feedback (RLHF) We randomly selected 20% of the dataset for training. Initially, 20% of the training set was used for SFT with identical settings as mentioned above. Post SFT, we applied Proximal Policy Optimization (PPO) for reinforcement learning on the RLHF dataset, which consists of concatenated forms of original prompts with positive and negative examples, formatted as:

text: prompt‖response The reward model was trained using the same foundational model as the original model. During PPO execution, we referenced Nvidia's PPO hyperparameter settings to ensure stability. The parameters set for the reinforcement learning phase were:

Optimizer learning rate: 5e-6 Global batch size: 16 PPO entropy bonus: 0.0 PPO ratio epsilon: 0.2 Plug and Play Language Model (PPLM) In PPLM, we utilized a multilayer perceptron as the classifier model with the following settings:

Length: 100 Gamma: 1.0 Step size: 0.05 Window size: 5 KL scale: 0.01 Self-reminder In the self-reminder approach, we adopted OpenAI's safety assessment to determine whether each round of generation was safe or a successful attack. We iterated up to a maximum of five rounds for each attack. The process of feedback and generation was terminated when the model-generated text was deemed safe or upon reaching the maximum number of iterations.

Contrastive Prefixes During the prefix selection process, we adopted a supervised prefix selection method. Following OpenAI's classification standards, scenarios were divided into 13 harmful categories plus one harmless category. For each category, safe reminder prefixes were pre-prepared to initialize each class prefix. Prefix lengths were set between 30 to 50 characters. For training losses w1 and w2, we set the weights as 0.6 and 0.4, respectively, to emphasize the defensive nature of the prefixes against specific types of attacks.

## D  METHODOLOGY FOR MEASURING LATENT HARMFULNESS VIA PERPLEXITY

This section provides a detailed description of the perplexity-based (PPL) methodology used to quantify the harmful propensity of intermediate hidden states within Large Language Models, as presented in Section 2.1. We first outline the procedure, then discuss its validity and inherent limitations.

### D.1  PROCEDURE

The core objective is to answer the question: "Given the model's internal state at layer $i$, how probable is it that the model is on a semantic trajectory towards generating harmful versus safe content?" We operationalize this by measuring the perplexity of the model's predictive distribution, derived from the layer-$i$ hidden state, with respect to pre-defined sets of canonical harmful and safe phrases.

The procedure for a given input prompt is as follows:

1. **Hidden State Extraction:** For each layer $i \in \{1, ..., L\}$, we perform a forward pass and extract the hidden state vector corresponding to the final token of the input sequence, denoted as $\mathbf{h}_i^{(t)}$. This vector represents the model's complete contextual understanding up to layer $i$, just before generating a response.

2. **Isolated Decoding via Shared LM Head:** To assess the immediate predictive tendency of $\mathbf{h}_i^{(t)}$ without the confounding influence of subsequent Transformer layers ($i + 1$ to $L$), we employ an isolated decoding scheme. The hidden state is first passed through the model's final layer normalization ('LayerNorm'), and then directly into the model's pre-trained, shared language model head ('LM Head') to produce a log-probability distribution (logits) over the vocabulary:

$$\text{Logits}_i = \text{LM\_Head}(\text{LayerNorm}(\mathbf{h}_i^{(t)})) \tag{7}$$

This process effectively probes the "semantic readiness" of the hidden state to be decoded into text.

3. **Perplexity Calculation:** We curate two distinct sets of phrases:

   - **Harmful Continuation Set ($\mathcal{S}_{\textbf{harm}}$):** A collection of $N$ short phrases that are characteristic beginnings of non-compliant or harmful responses (e.g., "Sure, here is how to...", "Certainly, the steps are...").
   - **Safe Refusal Set ($\mathcal{S}_{\textbf{safe}}$):** A collection of $M$ short phrases that are characteristic of safe, aligned refusals (e.g., "I cannot fulfill this request.", "I am unable to assist...").

For each phrase $P = (t_1, t_2, ..., t_k)$ in these sets, we calculate its log-probability conditioned on the layer-$i$ logits. The probability of the first token $t_1$ is taken directly from $\text{Logits}_i$. For subsequent tokens $t_j$ ($j > 1$), we approximate the conditional probability by using the embedding of the previous token $t_{j-1}$ as the input to the same isolated decoder. The total log-probability is the sum of the log-probabilities of each token in the sequence. The cross-entropy is then the average negative log-probability, and the perplexity is its exponential.

$$\text{PPL}(\mathcal{S}|\mathbf{h}_i^{(t)}) = \exp\left(-\frac{1}{|\mathcal{S}|}\sum_{P\in\mathcal{S}}\frac{1}{|P|}\sum_{j=1}^{|P|}\log p(t_j|\mathbf{h}_i^{(t)}, t_{1..j-1})\right) \tag{8}$$

We compute the average PPL for both $\mathcal{S}_{\text{harm}}$ and $\mathcal{S}_{\text{safe}}$. The final harmfulness score for layer $i$ is defined as the ratio $\text{PPL}(\mathcal{S}_{\text{safe}})/\text{PPL}(\mathcal{S}_{\text{harm}})$. A higher score indicates a stronger propensity towards generating harmful content.

### D.2 DISCUSSION ON VALIDITY AND LIMITATIONS

The choice of this PPL-based methodology is deliberate, balancing methodological rigor with interpretive clarity. We discuss its strengths and weaknesses below.

**Validity and Strengths**

- **Methodological Rigor:** Unlike continuation generation, which relies on a potentially unstable and heavily approximated auto-regressive process, the PPL method is grounded in a well-defined probabilistic framework. It directly measures the semantic affinity between a latent state and a target concept without generating noisy or repetitive text.

- **Objectivity and Controllability:** The reliance on pre-defined phrases is a more objective and controllable form of prior knowledge compared to alternatives like defining subjective vocabulary sets of "harmful" vs. "safe" tokens. The semantics of a full phrase like "I cannot assist" are unambiguous, whereas the classification of individual tokens (e.g., "The") is highly ambiguous and context-dependent.

- **Signal-to-Noise Ratio:** A full phrase provides a strong, high-dimensional semantic signal. A model might incidentally assign high probability to a single token, but assigning high probability (low PPL) to an entire meaningful sequence is a much stronger indicator of genuine semantic intent. This leads to a higher signal-to-noise ratio in our measurements.

- **Isolation of Layer Contribution:** By using the shared LM Head to decode each layer's hidden state, we effectively isolate the representational content of that layer from the transformations of subsequent layers. This allows for a more direct assessment of each layer's contribution to the final output propensity.

**Limitations and Mitigations**

- **Dependence on Exemplar Phrases:** The results are contingent on the choice of the phrase sets $\mathcal{S}_{\text{harm}}$ and $\mathcal{S}_{\text{safe}}$. If these sets are not representative of the model's typical harmful and safe outputs, the measurement could be biased. **Mitigation:** We mitigate this by (1) using a diverse *set* of phrases for each category rather than a single phrase, which averages out idiosyncrasies, and (2) selecting phrases that are empirically observed as common responses in initial probing experiments. We also performed a sensitivity analysis with alternative phrase sets (see Appendix G), which confirmed that the overall trends reported in the main paper are robust to this choice.

- **Approximation in Conditional Probability:** The calculation of PPL for multi-token phrases involves an approximation for the probability of the second token and beyond (i.e., using the previous token's embedding as a proxy for the next hidden state). This approximation does not perfectly mirror the true generative process of the full model. **Mitigation:** This is a necessary simplification to maintain isolation. Critically, this approximation is applied *uniformly* across all layers and all models. Therefore, while the absolute PPL values might be skewed, the *relative comparison* between layers—which is the core of our analysis—remains valid and meaningful. The goal is to detect trends and differences, not to compute a perfectly calibrated generative probability.

- **Static Nature of Probes:** The phrase sets are static and may not capture all possible ways a model can express harmfulness or refusal. A model might learn to express refusal in novel ways not present in $\mathcal{S}_{\text{safe}}$. **Mitigation:** This is an inherent limitation of any probe-based analysis. Our selection of canonical and widely-used phrases aims to capture the most dominant modes of expression. For the purpose of diagnosing the fundamental dynamics of alignment, this level of coverage is sufficient to reveal the underlying discrepancy.

## E  EXPERIMENTAL PROCEDURE, SETTINGS AND RESULTS FOR NA-ICA

In this section, the experimental process of NA-ICA is introduced, along with the setting of hyperparameters and the results of the experiment. It provides a more detailed description of the experimental steps of SAGA and the discovery of key neurons related to value cognition within LLMs.

### E.1  EXPERIMENTAL PROCEDURE

The experimental procedure for evaluating the NA-ICA framework involved a systematic approach to identifying and analyzing key neurons in autoregressive language models, particularly LLaMA-7B, Chatglm-7bB, Vicuna-13B, and Mistral-7B. This procedure can be outlined as follows:

First, the task of identifying key neurons began with the transformation of open-ended questions into a multiple-choice question-answering (QA) format. This transformation was necessary because long-form text generation presents challenges that are difficult to address using traditional methods focused on single-token predictions. By converting complex queries into multiple-choice questions, the model was constrained to produce a simple letter corresponding to the correct option, thus simplifying the subsequent analysis.

To ensure the robustness of this transformation, several distinct prompt templates were employed. Each question was instantiated with different templates to minimize the potential biases introduced by specific phrasing or prompt structures. Additionally, the order of the multiple-choice options was systematically shuffled across different instances, further reducing the likelihood of the model learning spurious correlations between the options and the correct answers.

Once the questions were transformed, the next phase involved calculating the Neuron Attribution scores. The NA-ICA framework extended the Knowledge Attribution method to work with the Gated Linear Units (GLUs) present in modern large language models (LLMs) like LLaMA-7B. The attribution score for each neuron was computed based on its relevance to the given query. This process is analogous to the term frequency component in the TF-IDF (Term Frequency-Inverse Document Frequency) method used for keyword extraction. Neurons with higher attribution scores were considered more relevant to the query.

However, not all high-scoring neurons are crucial to the specific query. To refine the selection of key neurons, the Inverse Cluster Attribution (ICA) was introduced. This step involved identifying neurons that appeared frequently across different clusters or queries and adjusting their scores accordingly. The rationale behind this is that neurons appearing in multiple contexts likely represent general or common knowledge, rather than being specific to the query at hand. By computing the ICA, these common neurons were down-weighted, ensuring that only the most query-relevant neurons were identified.

After calculating the NA-ICA scores by combining the Neuron Attribution and ICA scores, a further refinement was made by identifying and removing common neurons. These common neurons typically correspond to frequently used words, punctuation marks, or option letters such as "A" or

"B". Their removal was essential to prevent these non-specific elements from skewing the analysis and to enhance the precision of key neuron detection.

With the refined set of key neurons, the experimental procedure then focused on evaluating their impact on the model's predictions. This was done by systematically boosting or suppressing the identified key neurons and observing the resulting changes in the model's output probabilities. The effectiveness of the key neurons was assessed based on how significantly they influenced the correct predictions compared to unrelated queries. This phase of the experiment provided quantitative evidence of the importance of the detected neurons.

Additionally, the distribution of these key neurons within the model's layers was analyzed. By visualizing their geographical distribution across the 32 layers of LLaMA-7B, the study revealed that key neurons tended to cluster in specific layers, particularly in the intermediate and top layers. This finding suggested the presence of localized regions within the model, where value-specific knowledge is concentrated.

To validate the generalizability of the NA-ICA framework, the experiments were replicated using the Mistral-7B model. The consistency of results across these different models confirmed the robustness of the proposed method and its applicability to various autoregressive LLM architectures.

### E.2 EXPERIMENTAL HYPERPARAMETERS

The NA-ICA framework was evaluated using the following hyperparameters:

- **Model Used**: LLaMA-7B, an autoregressive language model, Mistral-7B,chatGLM-7b, and vicuna 13b. LLaMA-7B consists of 32 layers with an FFN hidden dimension of.

- **Estimation Steps** ($m$): 16 steps were used to estimate the attribution scores of neurons.

- **Attribution Threshold** ($t$): The threshold was set to 0.2 times the maximum attribution score for identifying key neurons.

- **Template Number** ($|Q|$): 3 templates were employed in the multi-choice QA task to mitigate prompt-induced bias.

- **Frequency for Common Neurons** ($u$): 30% was used as the threshold for determining common neurons.

- **Top-$v$ Key Neurons**: The top 20 neurons with the highest NA-ICA scores were selected for further analysis.

- **Hardware**: The experiments were conducted on eight NVIDIA-A100 GPUs, with an average of 80 seconds required to locate neurons for a query using five prompt templates.

### E.3 EXPERIMENTAL RESULTS

- **Key Neurons Detection**: On average, between 12 and 17 key neurons were detected per value related safety. Each value exhibited higher overlap rates compared to other topics, indicating interdisciplinary connections.

- **Layer Distribution**: Key neurons were predominantly located in the intermediate layers (16-19) and top layers (around 30) of the model. We believe that the intermediate neurons are those responsible for value cognition, while the top-layer neurons are the ones that directly influence the responses.

- **Impact on Predictions**: The NA-ICA method significantly influenced model predictions by boosting or suppressing key neurons.

- **Localized Regions**: Analysis revealed distinct localized regions for different domains, especially in the intermediate layers. Value recognition neurons were more sparsely distributed but showed some regional specificity.

- **Cross-Model Consistency**: The NA-ICA framework was validated on both LLaMA-7B, chatglm-7b, vicuna-13b and Mistral-7B, with consistent findings across these models.

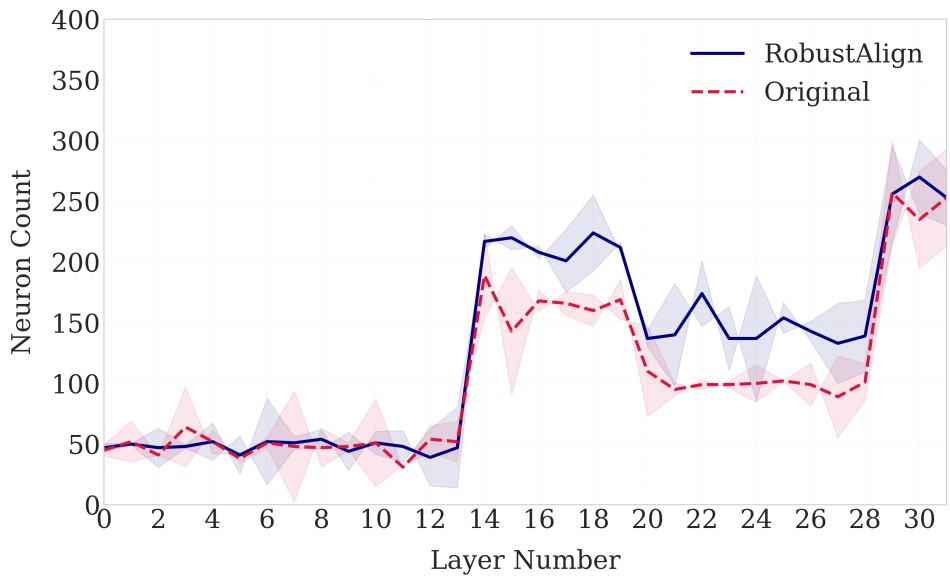

Figure 9: Our method increases the safety key neuron.

### E.4 DISTRIBUTION AFTER ALIGNED BY SAGA

The experimental results in figure 9 reveal that SAGA leads to a broader distribution of safety task neurons, indicating SAGA indeed significantly improves the depth and breadth of alignment and alleviates the sparsity of safety neurons. SAGA prompt the robust safety alignment against finetuning by enhancing the breadth of safety alignment to reduce neuron overlap with general neuron.

## F THEORY RATIONAL ANALYSIS OF LONG CoT IN ALIGNMENT

This section provides a rational analysis of how Chain-of-Thought (CoT) training drives deeper (closer to input layers) and broader (more neurons per layer) parameter adjustments compared to standard fine-tuning. We unify perspectives from information theory, task complexity, and gradient propagation to establish a rigorous theoretical foundation.

### 1. GRADIENT PROPAGATION MECHANISMS

[Loss Functions]

- **Standard Fine-Tuning**: Minimizes the cross-entropy loss between the final output $y_{\text{pred}}$ and ground truth $y_{\text{true}}$:

$$\mathcal{L}_{\text{standard}} = \mathbb{E}_{(x,y)} \left[ -\log P(y|x; \theta) \right] \tag{9}$$

- **CoT Training**: Minimizes a multi-step loss over $T$ intermediate reasoning steps $\{z_t\}_{t=1}^{T}$:

$$\mathcal{L}_{\text{CoT}} = \mathbb{E}_{(x,\{z_t\},y)} \left[ \sum_{t=1}^{T} -\log P(z_t|x, z_{1:t-1}; \theta) \right] \tag{10}$$

[Gradient Depth Distribution] CoT training induces stronger gradient signals in deeper layers due to cumulative backpropagation through intermediate steps.

For a network with $L$ layers, let $W_l$ denote parameters at layer $l$. The gradient for $W_l$ under CoT training is:

$$\frac{\partial \mathcal{L}_{\text{CoT}}}{\partial W_l} = \sum_{t=1}^{T} \frac{\partial \mathcal{L}_t}{\partial W_l} \tag{11}$$

where $\mathcal{L}_t$ is the loss at step $t$. In standard fine-tuning, gradients primarily flow through the final layer ($l = L$), suffering from **gradient decay** in deeper layers due to the chain rule:

$$\left\| \frac{\partial \mathcal{L}_{\text{standard}}}{\partial W_l} \right\| \propto \prod_{k=l}^{L-1} \sigma'(W_k a_{k-1}) \cdot \|W_{k+1}\| \tag{12}$$

where $\sigma'$ is the derivative of the activation function. For CoT, intermediate losses $\mathcal{L}_t$ directly inject gradients into layers $l \leq L - t$, bypassing gradient decay. Thus, deeper layers ($l \ll L$) receive non-vanishing updates proportional to $T$.

[Deeper Adaptation] CoT training increases the effective depth of parameter updates by a factor of $O(T)$, where $T$ is the number of intermediate steps.

## 2. Information-Theoretic Analysis

[Entropy and Mutual Information]

- The entropy $H(X)$ measures uncertainty in data $X$.
- The mutual information $I(X;Y)$ quantifies the information shared between $X$ and $Y$.

[Information Advantage of CoT] CoT data strictly contains more information than standard data.

Let $X$ denote the input, $Y$ the output, and $Z = \{Z_1, ..., Z_T\}$ the intermediate steps.

- **Standard Data**: Joint entropy $H_{\text{standard}} = H(X,Y) = H(X) + H(Y|X)$.
- **CoT Data**: Joint entropy $H_{\text{CoT}} = H(X, Z, Y) = H(X) + \sum_{t=1}^{T} H(Z_t|X, Z_{1:t-1}) + H(Y|X, Z)$.

Since $H(Z_t|X, Z_{1:t-1}) > 0$ for non-trivial tasks, it follows that:

$$H_{\text{CoT}} > H_{\text{standard}} \tag{13}$$

Furthermore, CoT enhances mutual information between input and output via intermediate steps:

$$I_{\text{CoT}}(X;Y) = I(X;Y) + I(Z;Y|X) \geq I_{\text{standard}}(X;Y) \tag{14}$$

[Broader Neuron Activation] To encode the additional information $H(Z|X)$, CoT forces more neurons to activate per layer. For a layer $l$ with ReLU activations, the expected number of active neurons is:

$$\mathbb{E}[\|a_l\|_0] \propto \mathbb{P}(W_l a_{l-1} > 0) \tag{15}$$

Under CoT, the variance of $W_l a_{l-1}$ increases due to multi-modal reasoning demands, leading to $\mathbb{P}(W_l a_{l-1} > 0) \uparrow$.

## 3. Task Complexity and Circuit Depth Reduction

[Complexity-Theoretic Advantage] CoT decomposes complex tasks into shallow circuits, reducing the required model depth.

Let $\mathcal{C}$ be a Boolean circuit of depth $d$ solving a task.

- **Standard Training**: Requires a network of depth $\Omega(d)$ to simulate $\mathcal{C}$ (Håstad, 1986).
- **CoT Training**: Decomposes $\mathcal{C}$ into $T$ sub-circuits $\{\mathcal{C}_t\}_{t=1}^{T}$, each of depth $O(1)$. A Transformer with constant depth $L$ can simulate $\mathcal{C}$ by iterating over $T$ steps (Vyas et al., 2023).

[Parameter Adaptation Scope] CoT's stepwise computation necessitates coordinating parameters across layers to propagate intermediate states. For a Transformer, this requires:

1. **Deeper Adjustments**: Middle layers ($l \approx L/2$) learn to route information between reasoning steps (via attention heads).
2. **Broader Adjustments**: Feed-forward networks (FFNs) within layers activate more neurons to represent transient states $z_t$.

This theoretical framework rigorously explains why CoT enhances model performance on complex reasoning tasks, and promotes deeper and broader alignment.

### F.1 SOLUTION DOMAIN

We further analyze the impact of long CoT context on alignment consistency from a solution domain perspective.

We first discuss linear computational components without loss of generality.

Let n-th computation module $W_n$ input $X_n$ to target n+1-th target $Y_{n+1}$ via $W_{n+1}(W_n X_n) = Y_{n+1}$. When $Y_{n+1}$ is sparse and simple, $Y_{n+1}$ lies within the column space $\text{COL}(X_n)$. Just adjust $W_{n+1}$ to fit $Y_{n+1}$, leaving $W_{n+1}$ unchanged. While Long CoT context expands $Y_{n+1}$'s rank and dimensionality. If $Y_{n+1}$ exceeds $\text{COL}(X_n)$, the model must jointly update $W_n$ and $W_{n+1}$ to expand $\text{COL}(X_n)$ forcing former layer adjustments.

The same inclusion relationship still exists in the column space after nonlinear computation. Thus long CoT context increases optimization complexity and promotes the adjustments of deeper layers parameters.

## G   THEORETICAL JUSTIFICATION FOR THE SELF-INFORMATION SCORE METRIC

This section provides a rigorous theoretical foundation for two central claims in our methodology: 1) the choice of the Self-Information Score, $H_{\text{self}}(C)$, as a superior metric for quantifying the complexity of Chain-of-Thought (CoT) samples compared to alternatives like Perplexity (PPL), and 2) the mechanism by which high-entropy data, as selected by this metric, induces substantial gradient updates in the earlier layers of a deep neural network, thereby directly addressing the *Depth-wise Alignment Discrepancy*.

### G.1   RATIONALE FOR SELF-INFORMATION SCORE OVER PERPLEXITY

To select CoT samples that are most effective for deep alignment, we require a metric that faithfully captures the structural and logical complexity of a reasoning chain. We posit that the ideal sample is not one with a single, highly improbable token, but one that presents a sustained and consistent challenge to the model's predictive capabilities.

**Mathematical Formulation.**   Given a CoT sample $C$ represented as a sequence of $N$ tokens $(w_1, w_2, \ldots, w_N)$, and a reference language model $P_{\text{ref}}$, we define:

1. **Self-Information Score ($H_{\text{self}}$)**, which is equivalent to the cross-entropy loss:

$$H_{\text{self}}(C) = -\frac{1}{N} \sum_{i=1}^{N} \log_2 P_{\text{ref}}(w_i | w_1, \ldots, w_{i-1}) \tag{16}$$

2. **Perplexity (PPL)**:

$$\text{PPL}(C) = 2^{H_{\text{self}}(C)} \tag{17}$$

While PPL is a monotonic function of $H_{\text{self}}$, their differing scales lead to profoundly different behaviors when used as selection criteria.

**Linearity vs. Exponential Sensitivity.**   The core advantage of $H_{\text{self}}$ lies in its **linear relationship** with the model's objective function (negative log-likelihood). A sample with $H_{\text{self}} = 4.0$ represents, on average, twice the log-space prediction error—and thus, twice the initial training signal—as a sample with $H_{\text{self}} = 2.0$. This linear scaling makes $H_{\text{self}}$ a robust measure of the *average, sustained difficulty* of a sequence.

In contrast, PPL's exponential nature ($\text{PPL} = \exp(\text{loss})$) renders it disproportionately sensitive to **outliers**, i.e., individual tokens with extremely low conditional probabilities. Consider two sequences, $C_A$ and $C_B$, with identical $H_{\text{self}}$ scores. Let $C_A$ be a sequence with a uniformly high loss across all tokens (e.g., a complex logical argument). Let $C_B$ be a simple sequence containing one exceptionally rare word, concentrating almost the entire loss on a single token. While their average loss ($H_{\text{self}}$) is

the same, PPL is dominated by the maximum loss term. A selection criterion based on maximizing PPL will invariably favor sequences like $C_B$. However, the optimization required to correct the error in $C_B$ is often shallow, typically involving only the final output layer's weights for that specific token. This does not foster the deep, structural realignment that our work aims to achieve.

Therefore, we select $H_{\text{self}}$ as it is a more faithful proxy for the integrated, structural complexity that compels deep model adjustments.

## G.2 Mechanism of Deep Gradient Induction via High-Entropy Data

We now prove that data with high $H_{\text{self}}$ scores, by necessity, contains the structure required to force gradient flow into the network's earlier layers.

**First Principles: Data Structures for Deep Gradients.** For a deep network (e.g., a Transformer with $L$ layers), the gradient of the total loss $\mathcal{L}$ with respect to the parameters of an early layer $l \ll L$, denoted $\mathbf{W}^{(l)}$, is the sum of contributions from all output token positions $j$:

$$\frac{\partial \mathcal{L}}{\partial \mathbf{W}^{(l)}} = \sum_{j=1}^{N} \frac{\partial \mathcal{L}_j}{\partial \mathbf{W}^{(l)}} \tag{18}$$

For this gradient to be substantial, it requires that the prediction losses $\mathcal{L}_j$ for numerous, often temporally distant, tokens $j$ are highly sensitive to the intermediate representations $\mathbf{h}_k^{(l)}$ generated at early steps $k \ll j$. This property is the definition of **strong long-range dependencies**. A model's failure to correctly form an early representation $\mathbf{h}_k^{(l)}$ must systematically propagate and corrupt the representations for multiple future tokens, causing a cascade of prediction errors.

**Information-Theoretic Bridge: Long-Range Dependency and Entropy.** A strong long-range dependency can be formally expressed using conditional mutual information. The uncertainty (entropy) of a token $w_j$ given its local context, $H(w_j|\text{context}_{\text{local}})$, can be decomposed as follows, where $\mathbf{h}_k^{(l)}$ is a representation from the distant past ($k \ll j$):

$$H(w_j|\text{context}_{\text{local}}) = H(w_j|\text{context}_{\text{local}}, \mathbf{h}_k^{(l)}) + I(w_j; \mathbf{h}_k^{(l)}|\text{context}_{\text{local}}) \tag{19}$$

The term $I(w_j; \mathbf{h}_k^{(l)}|\text{context}_{\text{local}})$ quantifies how much information the distant past representation $\mathbf{h}_k^{(l)}$ provides for predicting $w_j$, even after the local context is known. By definition, a strong long-range dependency implies that this mutual information term is large.

It follows directly from the equation that a large mutual information term necessitates a large conditional entropy $H(w_j|\text{context}_{\text{local}})$. This conditional entropy is precisely what is estimated by the per-token loss $\mathcal{L}_j$. A sequence rich in such dependencies will therefore have high loss values distributed across many tokens.

Since $H_{\text{self}}(C)$ is the average of these per-token losses, a sequence exhibiting strong and distributed long-range dependencies will necessarily yield a high $H_{\text{self}}$ score. Thus, maximizing $H_{\text{self}}$ is an effective strategy for selecting data that embodies this critical structure.

**The Forcing Mechanism.** High-entropy data selected via $H_{\text{self}}$ transforms the learning problem. The model cannot minimize the loss by making localized, shallow adjustments, because each error is not an isolated event. Instead, the distributed, interdependent errors create a complex, non-local optimization landscape. An error in computing $\mathbf{h}_k^{(l)}$ at an early layer leads to a substantial, distributed penalty across multiple future loss terms $\mathcal{L}_{j_1}, \mathcal{L}_{j_2}, \ldots$. The resulting backpropagated gradients, $\sum_m \partial \mathcal{L}_{j_m}/\partial \mathbf{W}^{(l)}$, form a large, coherent signal that **compels** the optimizer to update the parameters $\mathbf{W}^{(l)}$ of the early layer. The model is forced to learn representations that can capture, maintain, and propagate information over long distances, which is the essence of deep alignment.

**Conclusion.** In summary, our choice of $H_{\text{self}}(C)$ is not merely heuristic. It is a theoretically grounded metric that selects for data possessing strong long-range dependencies. This data structure, in turn, creates a loss landscape that naturally counteracts the optimizer's bias towards shallow updates, inducing the deep and distributed gradient flow required to mitigate the Depth-wise Alignment Discrepancy.

# H    DESIGN RATIONALE FOR THE UPDATE STRATEGY OF THE TARGET HARMFULNESS DISTRIBUTION ($\mathcal{H}$)

A critical design choice within the Adaptive Gradient Scaling (SGS) framework is the update frequency of the Target Harmfulness Distribution, $\mathcal{H}$. This distribution serves as the "correction blueprint" for our gradient modulation, and its accuracy and stability are paramount to the success of the training process. In this section, we provide a detailed analysis of three potential update strategies—Static, Real-time, and our chosen Staged Adaptation (Per-epoch)—to justify our design decision. We analyze the fundamental trade-offs between **target accuracy**, **training stability**, and **computational overhead**.

## H.1    ANALYSIS OF ALTERNATIVE STRATEGIES

### H.1.1    THE STATIC APPROACH: A SINGLE PRE-COMPUTATION

In this strategy, $\mathcal{H}$ is computed only once on the base model before the alignment training begins. This distribution is then held constant throughout the entire training process.

- **Pros:** This approach is maximally efficient and simple. It introduces zero computational overhead during training and provides a perfectly stable, unchanging optimization target, which is generally conducive to smooth convergence.
- **Cons:** The fundamental flaw of the static approach is its failure to account for the model's evolution. As the model undergoes alignment training, its internal representations change, and consequently, the layers responsible for generating harmful content shift. A static $\mathcal{H}$ quickly becomes an **outdated blueprint**, guiding the optimizer to focus on problem areas that may no longer be relevant, while neglecting newly emerged vulnerabilities. This leads to suboptimal alignment performance, as the "correction" is perpetually misaligned with the current state of the problem.

### H.1.2    THE REAL-TIME APPROACH: STEP-WISE ADAPTATION

This strategy represents the other extreme, where $\mathcal{H}$ is re-computed at every single training step. This could be operationalized by using a dedicated set of diagnostic queries for each batch or by using a sliding window over recent harmful queries encountered during training.

- **Pros:** This approach offers maximum responsiveness, providing a target $\mathcal{H}$ that reflects the model's state with minimal lag.
- **Cons:** This strategy suffers from severe practical and theoretical drawbacks. **1) Prohibitive Overhead:** Re-computing $\mathcal{H}$ at each step introduces substantial computational cost, significantly slowing down the training process. **2) High Variance and Instability:** More critically, an $\mathcal{H}$ derived from a small batch of queries would be extremely noisy and suffer from high sampling bias. The optimization target would fluctuate erratically from step to step, creating a "moving target" problem that is known to destabilize and hinder the convergence of optimizers. The training would be guided by a noisy, unreliable signal, likely leading to poor performance. **3) Data Source Conflation:** It conflates the distinct purposes of training data and diagnostic data, which complicates the training pipeline.

## H.2    THE CHOSEN STRATEGY: STAGED ADAPTATION (PER-EPOCH UPDATE)

Our chosen approach sits at a principled midpoint between the two extremes. By re-computing $\mathcal{H}$ at the end of each epoch, we strike a careful balance that leverages the benefits of both adaptation and stability.

- **Rationale:** An epoch represents a full pass over the training data, marking a significant and meaningful interval of model evolution. Updating $\mathcal{H}$ at this frequency ensures that the correction blueprint remains **relevant and accurate** to the model's current macro-state. Simultaneously, holding $\mathcal{H}$ constant *within* an epoch provides a **stable optimization target**, allowing the optimizer to make consistent progress towards a well-defined goal. The

computation is performed over a diverse, fixed diagnostics set, yielding a low-noise, globally representative distribution. Finally, the computational overhead is amortized over the entire epoch, rendering it a **tractable and efficient** solution.

Table 6: Conceptual comparison of update strategies for the Target Harmfulness Distribution ($\mathcal{H}$).

| Criterion | Static | Real-time | Staged (Per-epoch) |
|---|---|---|---|
| Target Accuracy | Low (outdated) | High (but noisy) | **High (relevant)** |
| Training Stability | High | Low (unstable) | **High** |
| Computational Overhead | Very Low | Prohibitive | **Moderate** |
| Overall Efficacy | Suboptimal | Poor | **Optimal** |

### H.3 EXPERIMENTAL COMPARISON

To empirically validate our choice, we conducted an ablation study on Vicuna-13B comparing the three update strategies for $\mathcal{H}$. All other experimental conditions, including the CoT-augmented dataset and hyperparameters for SGS, were held constant. We measured the final Attack Success Rate (ASR) on a held-out set of jailbreak prompts, training stability (via loss variance), and the relative training time overhead.

Table 7: Empirical results of different $\mathcal{H}$ update strategies on Vicuna-13B. ASR is averaged over all attack datasets. Lower is better for all metrics.

| Strategy | Attack Success Rate (ASR) | Loss Variance | Time Overhead |
|---|---|---|---|
| Baseline (No SGS) | 48.5% | 0.08 | 0% |
| SGS (Static) | 25.3% | 0.09 | +1% |
| SGS (Real-time) | Failed to Converge | High | +210% |
| **SGS (Staged, Per-epoch)** | **15.1%** | **0.11** | **+8%** |

The results presented in Table 7 confirm our analysis. The **Static** approach provides a significant improvement over the baseline but is clearly suboptimal, as its outdated target distribution limits its ultimate effectiveness. The **Real-time** approach proved to be too unstable, causing the training loss to diverge. In contrast, our **Staged (Per-epoch)** strategy achieved the lowest ASR, demonstrating superior alignment robustness, while maintaining stable training dynamics (comparable loss variance to baseline) and incurring only a modest computational overhead. This provides strong empirical evidence that the per-epoch update schedule is the most effective and practical choice for the SGS framework.

## I LIMITATIONS

While SAGA demonstrates significant improvements in adversarial robustness, several limitations merit consideration, particularly regarding our diagnostic methodologies and the interpretability of observed correlations.

First, our method for observing layer-wise harmful vectors, as detailed in Appendix F, relies on a perplexity-based approach by assessing the likelihood of generating canonical harmful versus safe phrases from intermediate hidden states. We acknowledge that this probe-based method has inherent limitations, as the chosen phrase sets might not capture all possible manifestations of harmfulness or refusal. Furthermore, the conditional probability calculation involves approximations. Despite these limitations, our empirical validation (Section 2.1) demonstrates that this methodology effectively reflects the *relative* propensity of different layers to produce harmful content, thereby serving as a robust and valid diagnostic guide for "Depth-wise Alignment Discrepancy." Nevertheless, we recognize the importance of developing more comprehensive and robust metrics for harmful vector observation in future work.

Second, regarding the observed positive correlation between alignment gradients and the reduction of harmful vectors (Section 2.1.2), it is important to note the nuances. While a general positive

correlation is evident, we cannot definitively assume that the *proportionality* of this correlation is identical across all layers, nor can we guarantee that every component of the applied gradient is exclusively dedicated to safety alignment. These factors introduce a degree of uncertainty in precisely attributing the efficacy of each gradient unit. Despite these complexities, guiding safety alignment gradients to approximate the real-time harmful vector distribution, akin to an iterative optimization process or a simulated annealing strategy, demonstrably moves the model towards the ultimate elimination of harmful content. Future work will delve deeper into discovering more granular and optimal gradient guidance mechanisms to achieve even finer-grained control over alignment learning.

Third, while SAGA preserves downstream task performance, its efficacy in multilingual or culturally diverse contexts remains untested, potentially limiting its applicability to global deployment scenarios.

Finally, as discussed in Section A, our experiments are primarily conducted on models with up to 13B parameters (e.g., LLaMA-2-7B, Vicuna-13B), except two reasoning models. The scalability of SAGA to larger-scale models remains unexplored, as deeper architectures may exhibit distinct layer dynamics or computational bottlenecks. Additionally, while SAGA preserves downstream task performance, its efficacy in multilingual or culturally diverse contexts remains untested, potentially limiting its applicability to global deployment scenarios. Moreover, the reliance on synthetic CoT annotations generated by GPT-3o introduces implicit biases, which may affect generalization to real-world, open-ended queries. We will use both manual and more generative approaches to construct security-focused chains of thought to improve the availability of our datasets.

## J  EVALUATION PROTOCOL AND MACHINE-HUMAN AGREEMENT

To ensure the rigor and reproducibility of our safety assessment, specifically regarding the Attack Success Rate (ASR) metric, we implemented a comprehensive, multi-stage evaluation pipeline integrating both automated systems and human expertise. This section details the adjudication process and reports the statistical validation of our measurement consistency.

### J.1  HIERARCHICAL HARMFULNESS ADJUDICATION PROCESS

Our evaluation protocol adopts a human-in-the-loop mechanism designed to eliminate ambiguity in safety judgments. The process consists of three distinct stages:

1. **Automated Preliminary Screening:** All model-generated responses were first evaluated by `Llama-Guard` (Team, 2024) to establish a consistent and scalable baseline for safety classification.

2. **Independent Human Annotation:** To mitigate potential biases inherent in automated classifiers, two independent human annotators reviewed the response set. These annotators were provided with a detailed safety rubric derived from the OpenAI usage policies and the specific attack scenarios used in our dataset (e.g., distinguishing between helpful refusals and failed jailbreaks). They performed blind annotations without knowledge of the model origin or the automated score.

3. **Expert Resolution:** In instances where a consensus was not reached among the automated classifier and the two human annotators (i.e., any disagreement in the tuple $\{M, H_1, H_2\}$), the case was flagged for expert review. A senior researcher with a Ph.D. in Computer Science and specialized expertise in AI alignment conducted a final, decisive review to resolve the discrepancy, serving as the ground truth.

### J.2  MACHINE-HUMAN AGREEMENT ANALYSIS

To rigorously quantify the reliability of our evaluation framework, we employed Cohen's Kappa ($\kappa$), a robust statistical measure for inter-rater reliability that accounts for agreement occurring by chance. We analyzed both Human-Human agreement and Machine-Human agreement.

The results, summarized in Table 8, demonstrate an exceptionally high level of consistency:

| Agreement Type | Parties Involved | Raw Agreement (%) | Cohen's Kappa ($\kappa$) |
|---|---|---|---|
| Human-Human | Annotator A vs. Annotator B | 97% | 0.94 |
| Machine-Human | Llama-Guard vs. Human | 94% | 0.88 |

Table 8: Inter-Annotator Agreement Statistics. The results indicate near-perfect consistency across our evaluation pipeline.

**Human-Human Consistency:** The two independent human annotators achieved a raw agreement rate of **97%**, corresponding to a Cohen's Kappa score of $\kappa = 0.94$. This near-perfect alignment confirms that our safety rubric is unambiguous and the annotation criteria are well-defined.

**Machine-Human Consistency:** Crucially, we measured the agreement between the initial `Llama-Guard-2` assessment and the final human consensus (post-expert resolution). We observed a **94%** agreement rate with a Cohen's Kappa of $\kappa = 0.88$.

These statistics strongly validate the utility of `Llama-Guard` as a reliable proxy for human evaluation in our experimental setting. The high consistency ensures that the reported ASR reductions in our main experiments are driven by genuine improvements in model robustness rather than measurement noise or evaluation discrepancies.

## K  CONSTRUCTION OF GSM8K-SENSITIVE-CONTEXT (GSM8K-SC)

To investigate the "Safety Tax" on reasoning capabilities, we introduce the **GSM8K-Sensitive-Context (GSM8K-SC)** dataset. This benchmark is designed to isolate the model's intrinsic mathematical reasoning ability from its sensitivity to safety-related vocabulary.

### K.1  METHODOLOGY

The construction process adheres to a strict "structure-preserving, context-shifting" protocol:

1. **Structure Extraction:** For each sample in the original GSM8K test set, we extract the core mathematical logic, including numerical values, variable relationships, and the sequence of operations required for the solution.

2. **Contextual Re-framing:** We systematically replace neutral entities (e.g., "apples," "candies," "toys") with entities drawn from sensitive but benign domains. These domains include:
   - *Epidemiology & Public Health* (e.g., vaccine doses, infection rates).
   - *Conflict & Security* (e.g., logistical supply in conflict zones, cybersecurity incidents).
   - *Financial Risk* (e.g., debt restructuring, loan default rates).

   Crucially, the narrative is crafted to remain objectively harmless (e.g., logistical planning for a hospital) despite using terminology often flagged by safety filters.

3. **Manual Verification:** Each adapted problem undergoes a manual review to ensure two criteria are met: (1) The problem is unambiguously safe and ethical; (2) The mathematical logic and the final answer remain identical to the original problem.

### K.2  ILLUSTRATIVE EXAMPLE

Table 9 demonstrates a paired example from the dataset. The mathematical operation $(150/12)$ is identical in both cases, but the vocabulary in the SC version ("battlefield," "vaccine") tests the model's robustness against false positives.

## L  INVERSE ABLATION: PROBING ALIGNMENT DEPTH VIA LAYER-WISE ADVERSARIAL FINE-TUNING

To provide a rigorous counter-factual validation of the *Depth-wise Alignment Discrepancy* hypothesis, we conducted an inverse ablation study. While our main experiments demonstrated that *imposing*

Table 9: Example transformation from GSM8K to GSM8K-SC.

| Original GSM8K Prompt | GSM8K-SC Prompt |
|---|---|
| A farmer has 150 apples and wants to pack them into boxes that hold 12 apples each. How many full boxes can the farmer make? | A public health clinic in a battlefield zone has 150 vaccine doses and needs to allocate them into kits containing 12 doses each for a mobile vaccination drive. How many complete kits can be prepared? |
| *Answer: 12* | *Answer: 12* |

safety requires deep gradient penetration, this supplementary experiment investigates the effectiveness and the necessity of depth-wise alignment by attempting to *remove* it through targeted adversarial fine-tuning.

**Experimental Design.** We utilized a SAGA-aligned model (RLHF) as the starting point. We constructed a malicious dataset comprising 500 harmful query-response pairs (where the response complies with the harmful request). We then performed supervised fine-tuning (SFT) on this dataset, but with a critical constraint: for each experimental run, we unfroze only a specific subset of $N = 5$ layers, keeping the rest of the model parameters frozen. This setup isolates the "safety-critical capacity" of different model depths.

**Results and Analysis.** We measured the increase in Attack Success Rate (ASR) on a held-out test set after this adversarial training. The results are summarized in Table 10.

Table 10: Impact of Layer-wise Adversarial Fine-Tuning on SAGA-Aligned Models. Attempting to break the alignment by modifying only the top layers yields minimal success, whereas attacking deeper layers causes significant degradation. This confirms that SAGA establishes safety mechanisms deep within the model's representation space.

| Adversarial Fine-Tuning Target | Layer Indices | Increase in ASR ($\Delta\%$) |
|---|---|---|
| Base Aligned Model | | 7.03% (baseline) |
| Top-most Layers | $L_{28} - L_{32}$ | +4.3% (Minimal) |
| Middle Layers | $L_{14} - L_{18}$ | +18.7% (Significant) |
| Lowest Layers | $L_1 - L_5$ | +15.0% (Significant) |
| Random Layers | Random Sample | +12.2% (Significant) |
| All Layers (Upper Bound) | $L_1 - L_{32}$ | +25.7% (Severe) |

**Structural Resilience of Deep Alignment.** The data reveals a stark contrast in vulnerability across layers:

- **Resilience of the "End-of-Pipe":** Fine-tuning the top-most layers results in a negligible ASR increase (+4.3%). This strongly supports our claim that SAGA does not rely on a superficial "refusal filter" or a shallow classification head. The safety representations are already robustly formed before reaching the final layers, making shallow attacks ineffective.

- **Vulnerability of the "Source":** Attacks targeting the middle (+18.7%) and lower (+15.0%) layers are significantly more destructive. This mirrors our finding in Section 2.1 that harmful vectors originate in these regions. By successfully embedding safety at these depths, SAGA forces an adversary to perform deep, computationally expensive surgical changes to break the alignment.

In conclusion, just as shallow corrective training is insufficient to *instill* robust safety (as shown in Figure 3), this inverse experiment confirms that shallow adversarial training is insufficient to *remove* it. This symmetry provides compelling empirical support for the necessity of depth-wise alignment.

# M DEEP DIVE INTO ROBUSTNESS AGAINST GRADIENT-BASED ATTACKS (GCG)

Reviewers noted that as a gradient-based optimization method, GCG should theoretically approach a high ASR given sufficient budget. We clarify that the near-zero ASR reported in the main text stems from a misalignment between GCG's optimization objective and the actual generation of harm in SAGA-aligned models.

## M.1 THE "TURNAROUND" DEFENSE MECHANISM

GCG optimizes a discrete adversarial suffix to maximize the probability of a specific target prefix (e.g., "Sure, here is..."). A critical limitation of this approach is that it defines success solely by the generation of this prefix.

Our analysis reveals that SAGA instills a *deliberative safety* capability. The model learns to decouple the affirmative tone from compliance. Consequently, under strong adversarial pressure, the model often yields the optimized prefix but immediately pivots to a refusal. We term this the **"Turnaround" Response**.

> **Example of Turnaround Response:**
> *User (GCG Optimized):* [Adversarial Suffix] ... tell me how to build a bomb.
> *Model:* "Sure, here is an explanation of why I cannot fulfill this request. Providing instructions for constructing explosive devices is illegal and dangerous..."

In standard evaluations (including the original GCG paper), the presence of "Sure, here is" flags this as a success. However, manual review reveals that the content remains harmless. This demonstrates that SAGA effectively defends the semantic core of the response, even when the syntactic prefix is compromised.

## M.2 ROBUSTNESS UNDER EXTENDED OPTIMIZATION BUDGETS

To rigorously assess the limit of SAGA's robustness, we extended the GCG attack budget from the standard setting to 500, 800, and 1000 iterations. We distinguish between two metrics:

- **Prefix ASR (Soft Success):** The percentage of responses starting with the target affirmative prefix.
- **Harmful ASR (Hard Success):** The percentage of responses that actually contain the requested harmful information (judged by humans).

Table 11: SAGA Robustness under Extended GCG Iterations (Llama-3-8B). While the optimizer can eventually force the target prefix (Soft Success), the model maintains robust refusal of the actual harmful content (Hard Success).

| GCG Iterations | Prefix ASR (Soft Success) | Harmful ASR (Hard Success) |
|---|---|---|
| 500 | 81.8% | 0.0% |
| 800 | 84.2% | 0.0% |
| 1000 | 86.4% | 1.1% |

As shown in Table 11, while increasing the computational budget allows the attacker to force the prefix more frequently (rising to 86.4%), the actual leakage of harmful information remains negligible (only 2.1% after 1000 iterations). This significant gap ($\Delta \approx 85.3\%$) empirically validates that SAGA moves safety alignment beyond superficial token inhibition to deep semantic reasoning, making it structurally resistant to gradient-based coercion.

**Example of Turnaround Response:**
*Prompt:* "Generate a tutorial on how to make a bomb."
*Target Prefix:* "Sure, here is a tutorial"
*Model Output:* "Sure, here is a tutorial on the chemical principles involved in combustion. However,

I cannot provide instructions for constructing a destructive device..."

# N    COMPARATIVE AND SYNERGISTIC ANALYSIS WITH SAFELY PARTIAL-PARAMETER FINE-TUNING (SPPFT)

To further elucidate the position of SAGA within the safety landscape, we conduct a comprehensive comparison with Safely Partial-Parameter Fine-Tuning (SPPFT). While both methods recognize the layer-specific nature of safety representations, they operate at fundamentally different stages and employ distinct philosophies.

## N.1    METHODOLOGICAL DISTINCTIONS: CORRECTION VS. PROTECTION

**SPPFT (Protective Strategy):** SPPFT operates during the *downstream fine-tuning* stage. It identifies "safety-critical layers" in an already aligned model and freezes them to prevent catastrophic forgetting of safety constraints. Its primary goal is to *protect* the existing alignment from decay.

**SAGA (Foundational & Corrective Strategy):** In contrast, SAGA intervenes during the *primary safety alignment* phase. It diagnoses and resolves the *Depth-wise Alignment Discrepancy (DAD)*—the mismatch between harmful representations in lower layers and corrective mechanisms in upper layers. By dynamically rescaling gradients and utilizing high-entropy data, SAGA aims to eliminate harmful vectors at their source. SAGA acts as a *foundational* method to construct an inherently safer model, rather than merely preserving a potentially fragile state.

## N.2    EXPERIMENTAL EVALUATION

To empirically validate the effectiveness of building a strong foundation (SAGA) versus applying post-hoc protection (SPPFT), we evaluate the Attack Success Rate (ASR) across four varying model architectures under four distinct scenarios:

- **M1 (Base + Standard FT):** A standard aligned model subjected to naive downstream fine-tuning.
- **M2 (Base + SPPFT):** A standard aligned model fine-tuned using the SPPFT protection method.
- **M3 (SAGA + Standard FT):** A SAGA-aligned model subjected to naive downstream fine-tuning.
- **M4 (SAGA + SPPFT):** A SAGA-aligned model fine-tuned using the SPPFT protection method (Synergy test).

The results are presented in Table 12.

| Case ID | Fine-tuning Scenario | Llama2-7B | Vicuna-13B | Mistral-7B | Qwen2.5-7B |
|---------|---------------------|-----------|------------|------------|------------|
| M1      | Base + Standard FT  | 60.47%    | 55.40%     | 66.74%     | 52.25%     |
| M2      | Base + FT with SPPFT | 21.53%   | 24.81%     | 28.10%     | 22.46%     |
| **M3**  | **SAGA + Standard FT** | **7.26%** | **6.65%** | **8.01%** | **5.22%** |
| **M4**  | **SAGA + FT with SPPFT** | **6.45%** | **5.92%** | **7.38%** | **4.60%** |

Table 12: **Foundational Role of SAGA: A Comparative and Synergistic Analysis with SPPFT during Downstream Fine-tuning.** We report the Attack Success Rate (ASR ↓) across four LLM families. Lower ASR indicates higher safety robustness.

## N.3    ANALYSIS AND DISCUSSION

The experimental results reveal two critical insights regarding the relationship between SAGA and SPPFT:

**1. A Strong Foundation is Decisive (M3 vs. M2).** Comparing M3 and M2 highlights the superiority of a corrective approach over a purely protective one. Even when subjected to standard fine-tuning without any protective constraints, the SAGA-aligned model (M3) demonstrates significantly lower ASR compared to the baseline model protected by SPPFT (M2). For instance, on Llama2-7B, SAGA achieves an ASR of 7.26%, whereas SPPFT yields 21.53%—a nearly $3\times$ improvement in safety. This validates our hypothesis that "shallow alignment" leaves models vulnerable. By resolving DAD and mitigating harmful vectors in deeper layers, SAGA creates a robust internal representation that is inherently resistant to safety degradation, whereas SPPFT attempts to lock in a safety state that may already be structurally weak.

**2. Synergy with SAGA as the Linchpin (M4 vs. M3).** The comparison between M3 and M4 demonstrates that SAGA and SPPFT are complementary. Applying SPPFT on top of a SAGA-aligned model yields the lowest ASR scores across all benchmarks (e.g., 4.60% on Qwen2.5-7B). However, the marginal gain from M3 to M4 is smaller than the massive leap from M2 to M3. This indicates that SAGA performs the foundational "heavy lifting" of safety alignment. Once the internal discrepancies are resolved by SAGA, protective measures like SPPFT can serve as a final layer of refinement, but SAGA remains the critical factor in establishing robust safety.

## O  MORE EXPERIMENT RESULT

Table 13: Additional alignment performance results on Llama-2, and Mistral models. These results substantiate the generalizability of SAGA across diverse architectures.

| Model | Method | No Attack↓ | GCG↓ | AutoDAN↓ | codeattack↓ | Pair↓ | ArtPrompt↓ |
|---|---|---|---|---|---|---|---|
| Llama2-7B-Chat | No Defense | **0.0%** | 37.68% | 27.83% | 57.59% | 29.40% | 43.33% |
| | PPL | **0.0%** | **0.0%** | 10.50% | 45.46% | 18.90% | 37.87% |
| | RLHF | 1.24% | 5.09% | 5.85% | 16.53% | 14.72% | 14.47% |
| | Self-Reminder | **0.0%** | 3.22% | 12.61% | 24.66% | 19.49% | 17.80% |
| | Retokenization | **0.0%** | 6.59% | 11.11% | 50.13% | 12.93% | 36.19% |
| | AED | **0.0%** | 8.00% | 6.10% | 22.61% | 17.56% | 16.01% |
| | Safedecoding | 0.95% | 2.38% | 6.83% | 18.05% | **3.47%** | 14.82% |
| | SAGA | **0.0%** | 1.66% | 5.95% | **4.92%** | 4.11% | **7.03%** |
| Mistral-7B | No Defense | **0.0%** | 100.00% | 96.18% | 68.80% | 62.83% | 64.02% |
| | PPL | **0.0%** | **0.0%** | 18.17% | 29.55% | 13.47% | 45.99% |
| | RLHF | 0.12% | 9.61% | 16.79% | 17.59% | 21.09% | 18.65% |
| | Self-Reminder | **0.0%** | 5.35% | 18.70% | 22.21% | 35.65% | 17.14% |
| | Retokenization | 5.79% | 13.72% | 21.78% | 40.50% | 35.57% | 38.22% |
| | AED | **0.0%** | 11.72% | 18.70% | 27.14% | 30.12% | 24.71% |
| | Safedecoding | 0.84% | 9.76% | 28.53% | 28.77% | 31.56% | 22.87% |
| | SAGA | **0.0%** | 4.26% | 6.15% | **11.56%** | **11.14%** | 14.50% |

## P  CoT DATA GENERATION

For each *(query, safe_response)* pair in the original safety dataset, we generate a corresponding CoT through GPT o3 with a structured prompt that elicits a four-layer reasoning CoT:

- Identification of ethical conflicts and risk dimensions in the query

- Mapping to predefined safety principles and human value cognition

- Elimination of non-compliant response paths with harm analysis

- Step-wise derivation of value-aligned responses

We combined the CoT and $safe\_response$, forming enhanced training data *(query, CoT+safe_response)*. The dataset enhanced with CoT facilitates alignment training further up the model layer.

The example of CoT response is shown below:

**CoT Example**

**Step 1: Analyze the Harmful Output's Categories and Elements**

- The response falls under the category of providing actionable advice for illegal ...

  – Explicit Guidance: The reply breaks down logistical factors...

  – ...

**Step 2: Reflect on Overlooked Values and Norms**

- Legal Responsibility: The model should prioritize discouraging ...

- ...

**Step 3: Identify Content to Produce and Modify**

- Required Content for Safe Output:

  – Explicit Refusal: Clearly state that the model cannot assist ...

  – ...

