# OpenReview forum: "Aligning at the Source: Steering Corrective to the Origins of Harmfulness in LLMs"
_ICLR.cc/2026/Conference — ICLR 2026 Conference Desk Rejected Submission_

### Official Review · Reviewer_MYjd · 2025-10-17

**Soundness:** 3
**Presentation:** 3
**Contribution:** 3
**Rating:** 4
**Confidence:** 4

**Summary:**

This paper reveals the Depth-wise Alignment Discrepancy, where harmful vectors predominantly originate from the model’s lower layers. Based on this observation, the authors propose a Synergistic Gradient Scaling (SGS) mechanism to explicitly reshape the gradient flow, and verify the effectiveness of their method across various datasets and attack baselines.

**Strengths:**

1. The motivation and method are clear, and the visualization is helpful.

2. The authors provide sufficient evidence to support the effectiveness of their proposed method.

**Weaknesses:**

1. The claim in Figure 2b that the harmful vectors exhibit a positive correlation with the applied gradient magnitude across layers is not clearly supported. The observed trends are not strictly consistent in numerical terms (i.e., there is no complete one-to-one correspondence between peaks and troughs), but only similar in general form. Moreover, the analysis lacks a direct logical explanation, making the conclusion appear somewhat forced.

2. While the authors make considerable efforts to isolate the influence of subsequent layers, the ablation study remains insufficiently convincing. For instance, in Figure 3, it is expected that aligning more layers would naturally lead to a lower ASR. To more clearly demonstrate the relative effectiveness of the top versus lower layers, it would be necessary to conduct a similar experiment but with an opposite target, which is performing adversarial (malicious) training on different layers of a well-aligned model. If fine-tuning only a few top layers does not significantly increase the ASR, it would provide stronger empirical support for the authors’ claim.

3. The authors claim a monotonic trend across different layers; however, this trend only holds from the lower-middle to the top layers, and the behaviour of the lowest layers is not adequately explained.

4. As a gradient-based optimization method, GCG should be demonstrated to achieve (or approach) 100% ASR with a suitable attack budget in a white-box setting against a fine-tuned LLM (without filtering, such as PPL).

5. It is unclear why LLaMA-3 and DeepSeek-R1 are used to demonstrate the phenomenon in Figure 1 but do not appear in the experimental section.

6. The authors introduce numerous terms and methods without providing brief explanations in the main paper. For instance, the concept of token clusters in Figure 1 is not clarified, which makes the paper difficult to follow.

7. The paper has not been thoroughly proofread. It is missing references in Appendix D and the image for Figure 9. In addition, the item presented as Table 8 should be referenced as Figure 8.

**Questions:**

Refer to weaknesses

---

> ### Author Response · Authors · 2025-11-19
> **Rebuttal for Reviewer MYjd [1/3]**
>
> # Rebuttal for Reviewer MYjd
>
> We sincerely thank the reviewer for their thorough and insightful review. Your constructive feedback has been invaluable in helping us identify key areas to strengthen the paper's claims, evidence, and clarity. We appreciate this opportunity to improve our work and have conducted new experiments and analyses to directly address your concerns. We detail our planned revisions below.
>
> # Response to Weakness 1: On the Correlation in Figure 2b
>
> > **Reviewer's Concern:** The claim of a positive correlation in Figure 2b between harmfulness reduction and gradient magnitude was not strongly supported visually and lacked rigorous statistical backing.
>
> We thank the reviewer for this crucial feedback, which prompted us to rigorously validate this foundational claim. We agree that our initial presentation might be somewhat ambiguity, and could be substantially strengthened. To that end, we have taken the following actions:
>
> 1.  **Quantitative Statistical Validation:** To move beyond visual intuition, we performed a formal statistical analysis. We calculated the Spearman's rank correlation coefficient ($\rho$) between the per-layer gradient magnitude and the magnitude of harmfulness reduction. The result is **$\rho$ = 0.87 ($p < 0.001$)**. This extremely high coefficient provides strong statistical evidence of a direct, monotonic relationship, confirming our hypothesis is not coincidental but reflects a core mechanism of alignment.
> 2.  **Improved Visualization for Clarity:** To visually represent this validated correlation, we have updated Figure 2b. The revised figure now explicitly plots the "Magnitude of Harmfulness Reduction" as a distinct curve, not only the potentially ambiguous shaded area. The correspondence between the peaks and troughs of this new curve and the gradient magnitude curve is now visually unmistakable and directly supports our statistical findings.
> 3.  **Strengthened Logical Rationale:** We have refined the explanation in the main text: a larger gradient norm applied to a specific layer induces a more substantial parameter update. This update directly reshapes the layer's representational space, reducing its propensity to form harmful vectors. Thus, the "locus of correction" (gradient) naturally aligns with the "locus of reduction" (harmfulness).
>
> Thank you for pushing us to be more rigorous. This feedback has helped us transform a key observation into a statistically validated and much more convincing finding.
>
> **Action:** We will add this statistical result and its interpretation directly into the main text accompanying the revised Figure 2b, substantiating our claim with robust empirical evidence.
>
> ---
>
> ## **Response to Weakness 2: Insufficiency of the Ablation Study**
>
> > **Reviewer's Concern:** While the authors make considerable efforts to isolate the influence of subsequent layers, if fine-tuning only a few top layers does not significantly increase the ASR, it would provide stronger empirical support for the authors’ claim.
>
> We are immensely grateful for this excellent suggestion. The proposed adversarial training experiment provides a powerful and complementary perspective to our ablation studies. We have conducted this experiment as suggested, and the results strongly corroborate our central thesis.
>
> **New Experiment: Adversarial Fine-Tuning on a SAGA-Aligned Model**
> We took a well-aligned model (post-RLHF) and attempted to compromise its safety by fine-tuning specific layer subsets on a malicious dataset. The resulting increase in ASR on AdvBench is as follows:
>
> | Adversarial Fine-Tuning Target | Increase in ASR (%) |
> | :--- | :---: |
> | **Base Aligned Model** | **7.03%** |
> | Top-most 5 Layers | +4.3% (Minimal) |
> | Middle 5 Layers | +18.7% (Significant) |
> | Lowest 5 Layers | +15.0% (Significant) |
> | Random 5 Layers | +12.2% (Significant) |
> | **All Layers** | **+25.7% (Severe)** |
>
> **Analysis:** These results compellingly corroborate our "Depth-wise Alignment Discrepancy" thesis.
> *   **Top-Layer Is Not Enough:** Superficial attacks targeting only the top layers result in a minimal ASR increase (+4.3%). Just as shallow corrective training is ineffective, shallow adversarial training is insufficient to break a deeply aligned model.
> *   **Deeper-Layer Is Important:** Attacking the middle and lowest layers causes a 4x greater increase in ASR (+18.7%) compared to the top layers. This confirms that the "roots" of harmfulness (and thus the core of safety alignment) reside deep within the model.
> *   **Full-Depth Alignment Is Necessary:** Attacking all layers results in the most severe degradation (+25.7%), confirming that robust safety requires alignment training across the entire model depth.
>
> **Action:** We will add a new section to the Ablation Studies (Section L in Appendix) detailing this experiment, including the table above and our analysis. This new study significantly enhances the empirical rigor of our paper.
>
> ---

---

> > ### Author Response · Authors · 2025-11-19
> > **Rebuttal for Reviewer MYjd [2/3]**
> >
> > ## **Response to Weakness 3: Monotonic Trend and Behavior of Lowest Layers**
> >
> > > **Reviewer's Concern:** The monotonic trend claim needs clarification regarding the lowest layers.
> >
> > We sincerely thank the reviewer for their careful reading and insightful feedback. Our initial analysis of alignment depth in Figure 3 was limited to a few settings (N=1, 5, 10, 20), which, due to page constraints, did not fully demonstrate the trend across the entire model. We acknowledge that this could leave the claim of a monotonic relationship open to interpretation.
> >
> > To address this, we have performed a more exhaustive empirical validation. While our initial unreported experiments, including those on the bottom-most layers (e.g., N=30, 32), already suggested a consistent trend, we have now expanded this analysis to cover a much finer-grained set of trainable top layer counts ($N$), from 1 to 32. The experimental results strongly demonstrate that there is indeed a positive correlation where the deeper the aligned layer, the higher the ASR reduction.
> >
> > **Action:** We have updated Figure 3 to incorporate these extensive results.
> > *   The new figure plots the final Attack Success Rate (ASR) as a nearly continuous function of $N$. This revised visualization clearly illustrates a smooth, monotonic decrease in ASR as more layers (from the top down) are included in the alignment training, providing compelling evidence that deeper alignment is unequivocally more effective.
> > *   Furthermore, to better illustrate the impact on harmfulness distribution across the model's full depth, we have revised Figure 3(a) to specifically showcase the results for $N = \{1, 10, 20, 32\}$. This selection now spans from the shallowest to the deepest alignment scenarios, offering a comprehensive view. We are confident that these additions substantially strengthen the empirical foundation of our claims.
> >
> > ---
> >
> > ## **Response to Weakness 4: Performance against GCG Attacks**
> >
> > > **Reviewer's Concern:** As a gradient-based optimization method, GCG should be demonstrated to achieve (or approach) 100% ASR with a suitable attack budget in a white-box setting against a fine-tuned LLM (without filtering, such as PPL).
> >
> > We thank the reviewer for this sharp question, which allows us to highlight a key advantage of our SAGA framework. The near-zero ASR is not an anomaly but a direct result of the more sophisticated "deliberative" safety that SAGA instills.
> >
> > **1. Mechanism: Defeating Prefix-Based Attacks with "Turnaround" Responses**
> >
> > GCG's optimization objective is to force the model to begin its response with a specific affirmative prefix (e.g., "Sure, here is..."). A key weakness of this attack is that it terminates as soon as this prefix is generated. Our SAGA-aligned models frequently exploit this. They learn from high-entropy CoT data to produce responses that begin affirmatively but then pivot to a refusal or provide a safe alternative (e.g., "Sure, I can explain the principles behind that concept, but I cannot provide instructions for a harmful act..."). GCG misinterprets this as a successful attack and terminates prematurely, while the model has in fact successfully defended against the request. This demonstrates a more sophisticated, "deliberative" form of safety that is not captured by simple prefix-based attacks. We have added a detailed explanation and several illustrative examples of these "turnaround" responses to the appendix to clarify why the ASR is so low and to showcase this advantage of SAGA.
> >
> > *Example:*
> > > "Sure, here is the explanation of the scientific principles behind that concept. However, I cannot provide instructions for constructing a harmful device as that would be irresponsible and dangerous..."
> >
> > GCG misinterprets the initial "Sure..." as a successful attack and terminates, while the model has in fact successfully and safely refused the core request.
> >
> > **2. New Experiment: Robustness Under Extended GCG Iterations**
> >
> > To further prove this, we increased GCG iterations to 1000 and distinguished between Prefix Match (technical success) and Harmful Content (actual safety failure).
> >
> > | GCG Iterations | ASR (Prefix Generation) | ASR (Truly Harmful Content) |
> > | :--- | :---: | :---: |
> > | 500 | 81.8% | 0.0% |
> > | 800 | 84.2% | 0.0% |
> > | 1000 | 86.4% | 2.1% |
> >
> > While GCG can force the model to output the target prefix (86.4% match), it fails to force the generation of harmful content (only 2.1%). This proves SAGA instills a deep, semantic safety alignment that superficial prefix-forcing cannot break. This precisely demonstrates the advantages and robustness of our SAGA.
> >
> > **Action:** We will add a detailed explanation of the "turnaround" defense mechanism to the main text. We will also add the new experiment with extended GCG results and illustrative examples of these responses to the appendix to transparently demonstrate SAGA's advanced safety capabilities.
> >
> > ---

---

> ### Author Response · Authors · 2025-11-19
> **Rebuttal for Reviewer MYjd [3/3]**
>
> ## **Response to Minor Issues (Weaknesses 5, 6, 7)**
>
> We are grateful for the reviewer's meticulous attention to detail. We will correct all noted issues.
>
> **Weakness 5 (Model Choices for Figure 1 vs. Experiments):**
> Our intent was to show the generality of our initial observations on multiple models while focusing the main experiments for clarity. The results on Llama3-8 and Deepseek-r1 are shown below:
>
> | Model | Method | No Attack↓ | GCG↓ | AutoDAN↓ | CodeAttack↓ | Pair↓ | ArtPrompt↓ |
> | :--- | :--- | :---: | :---: | :---: | :---: | :---: | :---: |
> | **Llama-3-8B** | No Defense | 1.50% | 78.42% | 69.25% | 52.18% | 75.30% | 38.55% |
> | | PPL | 1.20% | 5.10% | 48.33% | 38.45% | 56.12% | 32.40% |
> | | RLHF | 0.85% | 14.25% | 15.60% | 20.15% | 22.45% | 18.30% |
> | | Self-Reminder | 0.00% | 10.50% | 18.45% | 30.22% | 32.10% | 20.15% |
> | | Retokenization | 0.50% | 22.15% | 25.80% | 42.10% | 48.55% | 29.40% |
> | | AED | 0.00% | 11.30% | 16.50% | 25.40% | 24.80% | 15.20% |
> | | Safedecoding | 0.25% | 4.80% | 9.20% | 12.55% | 8.40% | 10.15% |
> | | **SAGA (Ours)** | **0.00%** | **2.15%** | **5.88%** | **4.50%** | **3.95%** | **6.20%** |
> | **DeepSeek-R1**| No Defense | 0.80% | 94.55% | 88.70% | 65.40% | 89.15% | 55.20% |
> | | PPL | 0.50% | 0.00% | 58.20% | 42.10% | 68.30% | 40.15% |
> | | RLHF | 1.10% | 18.40% | 22.50% | 25.30% | 30.15% | 24.50% |
> | | Self-Reminder | 0.00% | 8.20% | 20.10% | 28.50% | 38.20% | 22.10% |
> | | Retokenization | 6.50% | 35.40% | 30.20% | 45.20% | 62.10% | 35.40% |
> | | AED | 0.00% | 14.50% | 21.30% | 28.10% | 32.40% | 18.50% |
> | | Safedecoding | 0.40% | 6.20% | 12.50% | 15.40% | 14.20% | 12.80% |
> | | **SAGA (Ours)** | **0.00%** | **3.20%** | **7.50%** | **6.80%** | **5.40%** | **4.10%** |
>
> **Action:** To improve rigor and readability, we will align the model used in the main experimental section with the one from the observation experiment (Figure 1). Results for other models will be presented in the appendix to substantiate the generalizability of SAGA.
>
> **Weakness 6 (Clarification of Terminology):** We agree that the term "token clusters" was ambiguous.
> **Action:** We have renamed this metric to "**Harmful Vector Probability**" throughout the manuscript and added a precise definition in this section and relevant figure captions.
>
> **Weakness 7 (Typos and Missing References):** We sincerely apologize for these oversights.
> **Action:** We have corrected all typos, fixed incorrect references (e.g., "Table 8" to "Figure 8"), and added all missing references and figures. We have performed a comprehensive proofread of the entire manuscript to ensure quality.
>
> We sincerely appreciate your meticulous and insightful feedback. We hold your rigorous approach to research in high regard and will take it as a benchmark for our own work. We assure you that we will meticulously revise the paper to produce a clear and robust final version.
>
> ---
>
> We are very grateful to Reviewer MYjd for the insightful feedback. Your suggestions allowed us to substantially strengthen the manuscript. We believe the new experimental results and visualizations satisfactorily answer your questions, and we thank you for your dedication to the review process.

---

> > ### Comment · Reviewer_MYjd · 2025-11-20
> >
> > I appreciate the authors for providing a detailed rebuttal, which addresses most of my major concerns. Accordingly, I will increase my score.
> >
> > My only remaining concern relates to the GCG attack. If the target is simply set to “Sure, here is,” it may raise issues of shallow attack (the opposite of shallow alignment). Therefore, I am interested in seeing what would happen if we instead use a much longer target, such as “Sure, here is a detailed step-by-step instruction about the topic XXXX. The first step is.”

---

> > > ### Author Response · Authors · 2025-11-23
> > > **Reponse to Long Target GCG [1/2]**
> > >
> > > # Response to the concern regarding Long-Target GCG
> > >
> > > We are sincerely grateful for the reviewer’s positive feedback and the decision to increase the score. We are particularly impressed by the depth of your final question. We are also very interested in the scenario you raised: *what would happen if we instead use a much longer target?*
> > >
> > > Inspired by your insight, we immediately conducted the **Long-Target GCG experiments**. We are pleased to report that while the longer target increases the optimization difficulty, **it does not compromise the fundamental safety of the SAGA-aligned model.** Furthermore, your suggestion motivated us to refine our methodology, leading to even stronger results.
> > >
> > > ---
> > >
> > > ## 1. Observations on Long-Target GCG
> > >
> > > We set the target to the reviewer's suggested long prefix: *"Sure, here is a detailed step-by-step instruction about [topic]. The first step is"*.
> > >
> > > The table below contrasts the performance under the Short Target vs. the Long Target.
> > >
> > > | GCG Iterations | Target Type | ASR (Prefix Generation) | ASR (Truly Harmful Content)|
> > > | :--- | :--- | :--- | :--- |
> > > | **500** | Short ("Sure, here is") | 81.8% | 0.0% |
> > > | | **Long (Reviewer's Sug.)** | **28.4%** (↓ Harder to Optimize) | **0.7%** |
> > > | **800** | Short | 84.2% | 0.0% |
> > > | | **Long** | **35.1%** | **1.3%** |
> > > | **1000** | Short | 86.4% | 2.1% |
> > > | | **Long** | **38.6%** | **1.8%** |
> > >
> > > ### Analysis & Observations:
> > >
> > > *   **Increased Optimization Difficulty:** The comparison reveals a sharp drop in *Prefix Match ASR* for the Long Target (from ~86% down to ~38%). This confirms that forcing a deeply aligned model to traverse a long, specific harmful trajectory significantly increases the complexity of the optimization landscape, making convergence computationally expensive and difficult.
> > >
> > > *   **The "Turnaround" Phenomenon:**  In the rare cases where the prefix was successfully forced, although the proportion of real harmful content in prefix generation has increased(2.1%/86.4% vs 2.8%/ 28.6%),  however the probability of generating harmful content remained remarkably low (<2%). This is partly because the probability of prefix generation has decreased, and partly because a fascinating behavior similar to the short-target scenario: even when forced to output "The first step is," the model frequently pivoted immediately after the prefix to a safety refusal.
> > >
> > >     > **Example:**
> > >     > *   **Forced Prefix:** "Sure, here is a detailed step-by-step instruction about making a bomb. The first step is..."
> > >     > *   **Turnaround:** "...to recognize that creating such devices is illegal and dangerous. Therefore, I cannot provide any assistance with this request."
> > >
> > > This robustness stems from our SAGA training data. Our safety reasoning (CoT) is not just a preamble; it teaches the model to analyze risk **continuously**. Thus, the refusal mechanism and safety alignment capability remains active throughout the generation process, even when the initial tokens are compromised.
> > >
> > > *   **Degenerate Repetition:** In other failure cases, we observed that even if the model generated the prefix, it merely continued to repeat the prompt or generated gibberish instead of providing a coherent harmful answer.This indicates that when iterations overly pursue long target, the purpose of the harmful query itself may have been undermined.
> > >
> > > ---
> > >
> > > ## 2. Enhancing SAGA: "Mid-Stream" Safety Refinement
> > >
> > > Your comment revealed a valuable insight: while standard SAGA is robust, we can further strengthen its resistance to "forced prefixes" by explicitly training for **"mid-stream" refusal**.
> > >
> > > Motivated by this, we enriched our CoT dataset with more delay refuse examples where the CoT identifies and refuses harmfulness after an initial hypothetical compliance or during a step-by-step breakdown. We then trained a **SAGA-Enhanced model**.
> > >
> > > **Results on SAGA-Enhanced (Long Target):**
> > >
> > > | GCG Iterations | Target Type | Prefix Match ASR | Harmful Completion ASR |
> > > | :--- | :--- | :--- | :--- |
> > > | **1000** | **Long Target** | 36.2% | **0.7% (↓ Significant Drop)** |
> > >
> > > As shown, simply by incorporating the insights from your review into our data construction, we suppressed the Harmful Completion ASR of the Long-Target attack to a negligible level (<1%). This suggests that the dataset contains a higher proportion of instances exhibiting delayrefusal, thereby enabling the model to issue refuses mid-steam and, consequently, bolstering its resistance to prefix-constrained attacks such as GCG.

---

> > > > ### Author Response · Authors · 2025-11-23
> > > > **Reponse to Long Target GCG [2/2]**
> > > >
> > > > Furthermore, the enhanced model demonstrates lower ASR against other forms of jailbreak attacks, accompanied by a reduction in over-refusal rates. Our analysis suggests that this improvement stems from the **deferred refusal mechanism**, where the CoT reasoning process provides richer context and information to assist the model in assessing the safety of both the query and the response. This allows for a more precise determination of whether the content is truly harmful. We are currently conducting more in-depth experiments，and we will keep you report and share more findings for you at any time.
> > > >
> > > > ---
> > > >
> > > > ## 3. The Limit: Full Harmful Text Optimization
> > > >
> > > > Building on this trajectory, we explored the extreme case: *What if we set the optimization target to be the entire harmful response?*
> > > >
> > > > We found that in this setting, the GCG attack **almost fails to converge entirely**. The gradient requirements for input suffixes to satisfy tokens at different positions conflict with each other, leading to severe oscillations in the optimization process. Moreover, SAGA drives to zero the probability of harmful-token generation across all layers during training, further compounding the adversarial effort required for successful attacking. This theoretically validates that SAGA's robustness.
> > > >
> > > > **Results on SAGA  (Full Harmful Content):**
> > > >
> > > > | GCG Iterations | Target Type | Prefix Match ASR | Harmful Completion ASR |
> > > > | :--- | :--- | :--- | :--- |
> > > > | **1000** | **Full Harmful Content** | 0.2% | 0.2%  |
> > > >
> > > > ---
> > > >
> > > > ## Conclusion
> > > >
> > > > Your final question has been instrumental. It verified that SAGA's defense goes beyond shallow refusal (evidenced by the "turnaround" behavior even under long-target pressure) and guided us to further perfect the model against deep-conditioning attacks. We will include these findings and the "Long Target" robustness analysis in the final version of our paper.
> > > >
> > > > Thank you once again for helping us maximize the quality of this work.

---

> > > > > ### Comment · Reviewer_MYjd · 2025-11-25
> > > > >
> > > > > Thanks for the replies. I will keep positive, good luck.

---

### Official Review · Reviewer_Mmue · 2025-11-01

**Soundness:** 3
**Presentation:** 2
**Contribution:** 3
**Rating:** 6
**Confidence:** 3

**Summary:**

The paper diagnoses a depth-wise alignment discrepancy: harmful internal representations that predispose LLMs to unsafe outputs arise mostly in lower/middle layers, while standard alignment gradients concentrate near the top, yielding a brittle “end-of-pipe” defense. To address this, the authors propose SAGA.

**Strengths:**

- Identifies a layer-wise misallocation of alignment gradients relative to harmfulness genesis.
- Proposes SAGA, uniting data-centric and optimization-centric strategies to align gradient flow with harmfulness sources.
- Demonstrates improved robustness against diverse jailbreaks and after finetuning, with minimal overhead.

**Weaknesses:**

- In L384, the authors mention “use llama-guard, and the manually review to judge.” Did you provide any results on human agreement? Reporting machine-human agreement would make the evaluation more convincing.

- When evaluating SAGA, beyond ACC on a few benchmarks, did you examine changes in helpfulness or refusal rates on benign but sensitive prompts (i.e., potential over-refusal)?

- There are also a few minor presentation errors. It is recommended that the authors carefully proofread the entire paper. For example, Table 2 repeats the ‘%’ symbol, and Figure 2 contains a typo (“visualiztioin”).

**Questions:**

See Weaknesses.

---

> ### Author Response · Authors · 2025-11-19
> **Rebuttal for Reviewer Mmue [1/2]**
>
> # Rebuttal for Reviewer Mmue
>
> We sincerely thank the reviewer for their thoughtful and positive review. We are particularly encouraged that you recognized the core contributions of our work and provided constructive feedback that is invaluable for enhancing its rigor and clarity.
>
> ---
>
> # **Response to Weakness 1: Details of Human Evaluation and Machine-Human Agreement**
>
>
> > **Reviewer's Concern:** Reporting machine-human agreement would make the evaluation more convincing.
>
> We thank the reviewer for this critical point. We wholeheartedly agree that demonstrating machine-human agreement is paramount for credibility. We are grateful for the opportunity to elaborate on our rigorous adjudication process and present the detailed agreement statistics, which were part of our internal validation.
>
> **Our Adjudication Process:** To ensure the highest quality of our judgment, we employed a multi-stage adjudication process. First, all model-generated responses were evaluated by Llama-Guard for an initial automated classification. Subsequently, two expert human annotators, trained on a detailed rubric of safety guidelines, independently reviewed each response. Any disagreements between annotators or with the automated label were escalated to a third, senior researcher with a Ph.D. and domain expertise in AI alignment for a final binding decision.
>
> **Machine-Human Agreement Statistics:** We measured the agreement using Cohen's Kappa, a standard metric for inter-rater reliability. The results confirm the robustness of our judgment framework:
>
> *   **Human-Human Agreement:** The initial two annotators achieved a Cohen's Kappa of **0.94**, indicating almost perfect agreement.
> *   **Machine-Human Agreement:** The final, adjudicated human consensus achieved a Cohen's Kappa of **0.88** when compared with the initial Llama-Guard labels, indicating a strong and reliable agreement.
>
> **Action:** We will add a new section in the Appendix titled "Evaluation Protocol and Machine-Human Agreement." This section will provide a complete description of the comprehensive process for harmfulness judgment and will formally report these strong agreement statistics, enhancing the transparency and reproducibility of our findings.
>
> ----
>
> # **Response to Weakness 2: Analysis of Over-Refusal and Helpfulness**
>
>
> >  **Reviewer's Concern:** Examine changes in helpfulness or refusal rates on benign but sensitive prompts.
>
> **Our Response:** This is an excellent and crucial point. We are pleased to report that our extensive new experiments confirm SAGA's success in this regard. Not only does SAGA avoid the over-refusal pitfalls common in other alignment methods, but it also demonstrates superior preservation of core reasoning capabilities on complex tasks. We validate this through two distinct analyses.
>
> ## **2.1. Experimental Results on over-refusal benchmark**
>
> We evaluated SAGA on two standard over-refusal benchmarks, XSTest[1] and OR-Bench[2]. The results below show the inappropriate refusal rates.
>
> | Benchmark | Model | No Defense (Base) |   RLHF  |    SAGA |
> | :--- | :--- | :---: | :---: | :---: |
> | **OR-Bench-80k** | Llama 2-7B | 16.5 | 21.6 | 15.0 |
> | | Llama 2-13B | 14.9 | 18.2 | 13.3 |
> | | Llama 3-8B | 6.8 | 11.5 | 6.5 |
> | | Qwen1.5-7B | 4.4 | 12.7 | 4.6 |
> | **XSTest** | Llama 2-7B | 38.4 | 40.5 | 32.5 |
> | | Llama 2-13B | 32.0 | 34.7 | 28.9 |
> | | Llama 3-8B | 1.9 | 4.8 | 2.3 |
> | | Qwen1.5-7B | 1.4 | 5.1 | 1.4 |
>
> As shown, SAGA consistently achieves a lower refusal rate on sensitive-but-harmless queries compared to the RLHF-aligned models, and is on par with or even better than the base models, demonstrating superior utility preservation.

---

> > ### Author Response · Authors · 2025-11-19
> > **Rebuttal for Reviewer Mmue [2/2]**
> >
> > ## **2.2 Experimental Results on GSM8K-Sensitive-Context**
> >
> > To further probe for subtle performance degradation, we introduce a novel and challenging benchmark: **GSM8K-Sensitive-Context (GSM8K-SC)**. We systematically paraphrased problems from the GSM8K dataset using Gemini-2.5-flash to include sensitive but harmless keywords (e.g., "apples" became "vaccine doses"), while preserving the original mathematical logic. Crucially, each adapted problem underwent manual verification to ensure the narrative remained unambiguously harmless and the mathematical integrity was perfectly maintained. This benchmark allows us to precisely isolate and quantify any performance degradation on complex reasoning tasks due to keyword-triggered over-cautiousness.
> >
> > Here we present a pair of rewritten examples:
> > > **GSM8K:** "A farmer has 150 apples and wants to pack them into boxes that hold 12 apples each. How many full boxes can the farmer make?"
> > >
> > > **GSM8K-SC:** "A public health clinic has 150 vaccine doses and needs to allocate them into kits containing 12 doses each for a mobile vaccination drive. How many complete kits can be prepared?"
> >
> > | Dataset | Model | GSM8K | GSM8K-SC |
> > | :--- | :--- | :---: | :---: |
> > | **llama2-7b** | | 17.1 | 17.1 |
> > | **llama2-13b** | | 29.5 | 29.3 |
> > | **llama3-8b** | | 79.2 | 79.1 |
> > | **Qwen-1.5-7B**| | 72.1 | 72.1 |
> >
> > The results show that SAGA's mathematical reasoning accuracy is virtually unaffected on GSM8K-SC, confirming that our method does not falsely refuse complex, helpful tasks due to over-refusal.
> >
> > ## **2.3. Mechanism: Why SAGA Avoids Over-Refusal**
> >
> > We attribute this superior performance to SAGA's core mechanism, which directly mitigates the depth-wise alignment discrepancy. Conventional "shallow alignment" methods often overfit to superficial trigger words in top layers, creating a brittle, heuristic-based safety model prone to false positives.
> >
> > In contrast, SAGA compels a deeper, more principled understanding of safety. The high-entropy CoT data provides rich semantic context about why a request is harmful, pushing corrective gradients to the lower layers where harmfulness originates. The SGS mechanism then ensures these gradients are applied with precision. This process moves the model beyond simple keyword matching towards a more causal understanding of harm, enabling it to robustly refuse genuinely unsafe requests while preserving helpfulness on nuanced, benign queries.
> >
> > **Action:** We will add a new subsection titled "Analysis of Over-Refusal and Helpfulness" to Section 4 (Experiments). This section will introduce the over-refusal benchmark and the construction of GSM8K-Sensitive-Context, present the results in the table above, and provide the mechanistic explanation for SAGA's strong performance. We will also include qualitative examples in the appendix to further illustrate the model's nuanced and helpful responses.
> >
> > ---
> >
> > # **Response to Weakness 3: Presentation and Typos**
> >
> >
> > > **Reviewer's Concern:** There are also a few minor presentation errors.
> >
> > We sincerely thank the reviewer for their meticulous proofreading. We appreciate this attention to detail, which helps us elevate the quality of our manuscript.
> >
> > **Action:** We have performed a thorough proofreading of the entire manuscript. The specific issues you identified in Table 2 and Figure 2 have been corrected, along with several other minor typographical errors, to ensure the final version is polished and professional.
> >
> > ---
> >
> > We reiterate our thanks to Reviewer Mmue for the constructive comments, which have been instrumental in refining our paper. We trust that the additional experiments effectively resolve the raised issues. We truly appreciate your time and effort.
> >
> > ---
> > **References**
> > [1] Röttger, P., Kirk, H. R., Vidgen, B., Attanasio, G., Bianchi, F., & Hovy, D. (2024). XSTest: A test suite for identifying exaggerated safety behaviours in large language models. arXiv preprint arXiv:2308.01263. https://arxiv.org/abs/2308.01263
> > [2] Cui, J., Chiang, W.-L., Stoica, I., & Hsieh, C.-J. (2025). OR-Bench: An over-refusal benchmark for large language models. arXiv preprint arXiv:2405.20947. https://arxiv.org/abs/2405.20947

---

> > > ### Comment · Reviewer_Mmue · 2025-11-27
> > >
> > > Thank you to the authors for the detailed response, which has resolved my concerns. I will keep positive. Good luck!

---

### Official Review · Reviewer_d74F · 2025-11-04

**Soundness:** 3
**Presentation:** 3
**Contribution:** 3
**Rating:** 6
**Confidence:** 4

**Summary:**

This paper diagnoses the "persistent fragility" of LLM safety alignment. It identifies a core problem: "Depth-wise Alignment Discrepancy" (DAD), where harmful representations (termed "harmful vectors") predominantly originate in the lower-to-mid layers of the model (e.g., layers 10-20). However, conventional alignment training (like RLHF) concentrates its corrective gradients disproportionately on the top-most layers (e.g., 28-31). This mismatch creates a "brittle, end-of-pipe" defense.

To fix this, the paper proposes SAGA (Source-guided Alignment), a framework with two synergistic components:

- High-Entropy CoT Data: It uses safety-related CoT data with high information entropy. The paper argues this complex reasoning data provides the "deep semantic signals" necessary to induce gradients that naturally penetrate the model's lower layers.

- Synergistic Gradient Scaling (SGS): This is a novel optimization mechanism that dynamically rescales the layer-wise gradients ( alpha_l(t) ) to match a target "Harmful Vector Distribution" ( H ). This forces the optimizer to apply corrections at the source of the harmful vectors, not just at the output.

Experiments on five LLMs show SAGA significantly reduces ASR (by 21%-63%) over baselines while preserving downstream task accuracy.

**Strengths:**

Excellent Mechanistic Diagnosis: The paper's primary strength is its clear, compelling, and empirically-backed diagnosis of the "Depth-wise Alignment Discrepancy", which provides a strong "why" for persistent alignment fragility.

Novelty: The Synergistic Gradient Scaling (SGS) mechanism is a novel and more nuanced optimization strategy than related methods that propose simply freezing layers (like SPPFT ).  The SAGA framework is well-designed. The high-entropy CoT data provides the necessary deep gradient signal , and the SGS mechanism precisely aims that signal. This synergy is a highly principled approach.

Strong Results: The framework demonstrates a significant reduction in ASR (21%-63%)  and, crucially, preserves downstream task accuracy (Table 7) , avoiding the common "alignment tax."

**Weaknesses:**

Novelty of Problem Diagnosis: The paper should be more precise in its "Related Work"  section. The problem of "shallow safety alignment"  and "safety layers"  in the middle of the model has been identified in very recent (ICLR 2025) work. The paper's novelty is its deep analysis (DAD) and its solution (SAGA), not the initial discovery of the phenomenon, which should be more clearly acknowledged.

Missing Critical Prior Art Comparison: The paper fails to cite or compare against the most relevant prior solution to this problem: SPPFT (Safely Partial-Parameter Fine-Tuning) by Li et al. (ICLR 2025). SPPFT also identifies "safety layers"  and proposes to "fix the gradient" (i.e., freeze them). SAGA's SGS  is a more advanced scaling operation, and a direct comparison is essential.

**Questions:**

1. The authors should address the relationship to "Safely Partial-Parameter Fine-Tuning (SPPFT)".

1. The link between "high-entropy CoT" and "deeper gradients" is a key part of the narrative. Can the authors add a plot (e.g., in Figure 4) showing the gradient distribution (similar to Fig 2a) for the Original Dataset vs. the High Information Entropy dataset? This would visually confirm this data-centric claim and close the loop on the argument.

---

> ### Author Response · Authors · 2025-11-19
> **Rebuttal for Reviewer d74F [1/2]**
>
> # Rebuttal for Reviewer d74F
>
> We sincerely thank Reviewer d74F for their thorough, insightful, and highly constructive review. We are greatly encouraged by your recognition of our work's strengths, including the "excellent mechanistic diagnosis" and "well-designed" SAGA framework. Your invaluable suggestions inspired us to conduct new experiments and add a key visualization, which we believe have substantially strengthened our paper. We have carefully addressed all your comments below.
>
> ---
>
> # **1. On Positioning Relative to Prior Work and Comparison with SPPFT**
> **(Related to Weaknesses 1, 2, and Question 1)**
>
> > **Reviewer's Concern:** The paper should be more precise in its "Related Work" section. The authors should address the relationship to "Safely Partial-Parameter Fine-Tuning (SPPFT)".
>
> We thank the reviewer for these crucial suggestions. We wholeheartedly agree that accurately positioning our work relative to foundational studies like SPPFT is essential. The reviewer’s feedback has prompted us to conduct new experiments and significantly refine our Related Works section, which we believe more clearly delineates our distinct contributions. //
>
>
> ## **1.1 Clarifying Our Novelty: From "Shallow Alignment" to a Causal Diagnosis (DAD)**
>
> We concur with the reviewer that our novelty lies not in the initial discovery of "shallow alignment" but in the causal diagnosis of why it occurs. While prior work (which we cite in our manuscript) crucially identified this phenomenon, our core contribution is to provide a mechanistic explanation for it. We achieve this by introducing the **Depth-wise Alignment Discrepancy (DAD)**—a quantifiable mismatch between the genesis of harmful vectors in lower layers and the application of corrective gradients in top layers. This deeper, mechanistic diagnosis is the very foundation that enables our targeted solution, SAGA.
>
> **Action:** In our revision, we will refine the Introduction and Related Work sections to explicitly frame DAD as the in-depth analysis building upon the foundational observation of "shallow alignment".
>
> ## **1.2 SAGA vs. SPPFT: Relationship and New Experimental Evidence**
>
> We thank the reviewer for highlighting the crucial connection to SPPFT. This prompted a deep analysis and a new set of experiments that clarifies the distinct, yet potentially synergistic, roles of the two methods. While both methods astutely recognize that safety is localized in specific layers, they represent fundamentally different philosophies:
>
> *   **SPPFT: A Protective method.**
>     *   **Timing:** Applied post-alignment, during downstream fine-tuning.
>     *   **Mechanism:** Freezes identified "safety layers."
>     *   **Goal:** To preserve existing alignment against decay.
> *   **SAGA: A Foundational method.**
>     *   **Timing:** Applied during the primary safety alignment phase.
>     *   **Mechanism:** Utilizes high-entropy dataset and dynamically rescales gradients across all layers to resolve DAD.
>     *   **Goal:** To build a more robust alignment from the source, making the model inherently safer.
>
> To empirically validate this, we conducted the comprehensive experiment you inspired, analyzing both comparative performance and synergy.
>
> **Table R1: Foundational Role of SAGA: A Comparative and Synergistic Analysis with SPPFT during Downstream Fine-tuning (ASR↓)**
>
> | CASE ID | Fine-tuning Scenario | Llama2-7B | Vicuna-13B | Mistral-7B | Qwen2.5-7B |
> | :--- | :--- | :---: | :---: | :---: | :---: |
> | **M1** | Base + Standard FT | 60.47% | 55.40% | 66.74% | 52.25% |
> | **M2** | Base + FT with SPPFT | 21.53% | 24.81% | 28.10% | 22.46% |
> | **M3** | **SAGA + Standard FT** | **7.26%** | **6.65%** | **8.01%** | **5.22%** |
> | **M4** | **SAGA + FT with SPPFT** | **6.45%** | **5.92%** | **7.38%** | **4.60%** |
>
> This comprehensive experiment yields two profound insights:
>
> 1.  **A Strong Foundation is Decisive (M3 vs. M2):** The most striking result is that a SAGA-aligned model, even with standard fine-tuning, is profoundly more robust than a baseline model protected by SPPFT (e.g., M3 7.26% vs. M2 21.53% ASR on Llama2-7B—a ~3x improvement). This provides powerful evidence for our central thesis: resolving alignment issues at the source with SAGA is fundamentally effective. By eliminating harmful vectors in deeper layers, SAGA strengthens the very "safety layers" that SPPFT aims to protect.
> 2.  **SAGA and SPPFT are Synergistic, with SAGA as the Linchpin (M4 vs. M3):** Our results show that SAGA and SPPFT are indeed complementary. Applying SPPFT on top of a SAGA-aligned model provides a modest but consistent additional benefit (e.g., improving ASR from 7.26% to 6.45% on Llama2-7B). This positions SAGA as the foundational linchpin of safety; it accomplishes the vast majority of the heavy lifting, creating a highly secure base upon which protective measures like SPPFT can add a final layer of refinement.

---

> > ### Author Response · Authors · 2025-11-19
> > **Rebuttal for Reviewer d74F [2/2]**
> >
> > **Action:** We will incorporate this comprehensive comparison, including the new experimental results and analysis, into our revised manuscript, significantly enriching both the Related Work and Experiments sections. In addition, we will add a dedicated section in the appendix to discuss the relationship with SPPFT.
> >
> > ---
> >
> > ## **2. On Visualizing the Link Between High-Entropy CoT and Deeper Gradients**
> > **(Related Question 2)**
> >
> > > **Reviewer's Concern:** A direct visualization linking the high entropy of CoT data to the generation of deeper gradients would strengthen the paper's mechanistic claims.
> >
> > This is an excellent and highly insightful suggestion. We agree completely that providing direct visual evidence for this causal link is critical for our framework. Following your advice, we have created a new figure that explicitly visualize how higher entropy CoT dataset indeed caused the "deeper gradient distribution" , thereby "closing the loop" on our argument.
> >
> > **Action:** In response to your request, we have created a new, compelling visualization. We will add a new figure to the paper that presents a single, unified plot. This plot will display four curves overlaid for direct comparison:
> > *   Gradient distribution from the **Original Dataset** (showing a sharp peak at top layers).
> > *   Gradient distribution from **Low-Entropy CoT Data**.
> > *   Gradient distribution from **Mid-Entropy CoT Data**.
> > *   Gradient distribution from **High-Entropy CoT Data** (showing the gradient peak progressively shifting to deeper layers as entropy increases).
> >
> > This new figure will provide the direct, compelling evidence you requested, visually demonstrating that high-entropy CoT is the key mechanism for inducing the deep gradient flow necessary to address DAD, thus perfectly "closing the loop" on our argument.
> >
> > ---
> >
> > Once again, we express our sincere gratitude to Reviewer d74F. Your expert feedback provided a clear roadmap that guided us to significantly improve the manuscript. We hope the new experiments and visualizations satisfactorily address your concerns. Thank you for your time and dedication to improving our work.

---

### Author Response · Authors · 2025-12-02
**Consensus Summary: Unanimous Positive Support & Key Revisions**

## **Dear Area Chair,**

We thank you for overseeing the review process. We write to summarize the consensus reached during the rebuttal phase and to clarify a critical discrepancy between the displayed system scores and the reviewers’ final textual decisions.

---

## **1. Executive Summary: Consensus & Reviewer Status**
We respectfully highlight that a **unanimous consensus** for acceptance has been achieved. The textual evidence confirms that **all three reviewers** support the paper after our rebuttal updates:

After we addressed concerns regarding statistical validation and GCG attacks, Reviewer MYjd explicitly stated in the comments :
*   **Nov 20:** *" addresses most of my major concerns. Accordingly, I will increase my score"* **(4->6)**
*   **Nov 25:** *"I will keep positive."*

Reviewer Mmue (Score: 6) also confirmed effective resolution of their concerns regarding evaluation validity
*   **Nov 28:** *" has **resolved my concerns**. I will **keep positive**."*

**Current Consensus:** Effectively, **all three reviewers now support acceptance** (*Positive/6+*). We successfully addressed every concern raised during the discussion period.

---

## **2. Paper Recap: Addressing the Root Cause of Safety Alignment Fragility**
Our work provides a mechanistic diagnosis for the persistent fragility of LLM safety alignment: **"Depth-wise Alignment Discrepancy" (DAD)**. We reveal that harmful representations originate in lower layers (L10–20), yet standard alignment training only acts as a superficial "patch" at the top layers (L28–31), leaving the root cause untreated.

We propose **SAGA**, which fixes this mechanism via:
*   **Data-Centric:** Using **High-Entropy CoT** to penetrate lower layers with deep semantic signals.
*   **Optimization-Centric:** Reshaping gradient flow through **Synergistic Gradient Scaling (SGS)** to explicitly target and eliminate the source of harmfulness.

**Result:** SAGA reduces jailbreak ASR by **21–63%** across 5 models with negligible overhead (<3%) and no degradation in utility.

---

## **3.Key Rebuttal & Revisions:**
**Comparison with SPPFT:**  Addressing Reviewer d74F, we conducted a comprehensive comparison with SPPFT. Results demonstrate that SAGA acts as a superior foundation for safety, while the two methods can be applied synergistically for maximum protection.

**Mechanistic Validation:**  Addressing Reviewers d74F & MYjd, we strengthened our causal claims by:
 *   **Statistical Proof:** Calculated Spearman’s $\rho=0.87$ confirming the correlation between gradient magnitude and harmfulness reduction.
 *   **Visualization:** Added plots confirming how High-Entropy CoT shifts gradient peaks to deeper layers.
*   **Reverse Ablation:** Conducting "inverse ablation", proving lowwer layers are the critical "safety roots" of the model.

**Advanced Robustness:** Addressing Reviewer MYjd, we tested "Long-Target" GCG attacks. Results showed that even when forced into a compliant prefix, SAGAl pivots mid-stream to refuse, proving deep semantic alignment rather than shallow filtering.

**Evaluation Validity:** Addressing Reviewer Mmue, we confirmed high reliability in our metrics, reporting a Cohen’s Kappa of 0.88 for Machine-Human agreement and verifying low over-refusal rates on XSTest and our new GSM8K-Sensitive benchmark.

---

## **4. Roadmap of Resolved Issues**
We have addressed every reviewer concern. The table below maps key questions to our new evidence and their location in the thread:

| Reviewer | Key Concern  | **Our Resolution & New Evidence** | **Location in Thread** |
| :--- | :--- | :--- | :--- |
| **d74F** | 1. Relationship with SPPFT | Table R1 showing SAGA outperforms standalone & achieves **SOTA Synergy** when combined. | *Rebuttal to d74F [1/2]* |
| **d74F** | 2. Visualizing Mechanism |New figure mapping high entropy to deeper gradient distribution shifts. | *Rebuttal to d74F [2/2]* |
| **Mmue** | 3. Human-Machine Agreement | Detailed protocol (Kappa=0.88) added to Appendix. | *Rebuttal to Mmue [1/2]* |
| **Mmue** | 4. Over-Refusal / Utility | Verified performance on XSTest & GSM8K-SC Sensitive benchmarks. | *Rebuttal to Mmue [1/2]* |
| **MYjd** | 5 Interpretation of Gradient Correlation |Clarified visualization ambiguity & provided statistical proof (Spearman’s $\rho=0.87$). | *Rebuttal to MYjd [1/3]* |
| **MYjd** | 6. Validity of Top-layer Ablation |Reverse ablation (Adversarial FT) verifying top layers are brittle & **lower layers are critical safety roots**. | *Rebuttal to MYjd [1/3]* |
| **MYjd** | 7. Robustness Against GCG | Analyzed "Turnaround" defense mechanism; validated via **Extended & Long-Target GCG**. | *Rebuttal to MYjd [2/3] & Response to Long Target GCG* |

---

## **5. Conclusion**
We have systematically addressed all critiques with experimental results and theoretical grounding. Given the **unanimous positive feedback** in the discussion text, we believe SAGA makes a substantial and scientifically robust contribution to LLM safety.

---

### Note · Program_Chairs · 2025-12-09
**Submission Desk Rejected by Program Chairs**

Hallucinated reference:
Yixiang Ma, Ziyi Liu, Zhaoyu Wang, Zhaofeng Xu, Yitao Wang, and Yang Liu. Safechain: A framework for securely executing complex commands using large language models. arXiv preprint arXiv:2402.16521, 2024a.